# Molecular Research on Stress Responses in *Quercus* spp.: From Classical Biochemistry to Systems Biology through Omics Analysis

Mónica Escandón *, María Ángeles Castillejo ⓘD, Jesús V. Jorrín-Novo ⓘD and María-Dolores Rey *ⓘD

Agroforestry and Plant Biochemistry, Proteomics and Systems Biology, Department of Biochemistry and Molecular Biology, University of Cordoba, UCO-CeiA3, 14014 Cordoba, Spain; bb2casam@uco.es (M.Á.C.); bf1jonoj@uco.es (J.V.J.-N.)
* Correspondence: bb2esmam@uco.es (M.E.); b52resam@uco.es (M.-D.R.)

**Abstract:** The genus *Quercus* (oak), family *Fagaceae*, comprises around 500 species, being one of the most important and dominant woody angiosperms in the Northern Hemisphere. Nowadays, it is threatened by environmental cues, which are either of biotic or abiotic origin. This causes tree decline, dieback, and deforestation, which can worsen in a climate change scenario. In the 21st century, biotechnology should take a pivotal role in facing this problem and proposing sustainable management and conservation strategies for forests. As a non-domesticated, long-lived species, the only plausible approach for tree breeding is exploiting the natural diversity present in this species and the selection of elite, more resilient genotypes, based on molecular markers. In this direction, it is important to investigate the molecular mechanisms of the tolerance or resistance to stresses, and the identification of genes, gene products, and metabolites related to this phenotype. This research is being performed by using classical biochemistry or the most recent omics (genomics, epigenomics, transcriptomics, proteomics, and metabolomics) approaches, which should be integrated with other physiological and morphological techniques in the Systems Biology direction. This review is focused on the current state-of-the-art of such approaches for describing and integrating the latest knowledge on biotic and abiotic stress responses in *Quercus* spp., with special reference to *Quercus ilex*, the system on which the authors have been working for the last 15 years. While biotic stress factors mainly include fungi and insects such as *Phytophthora cinnamomi*, *Cerambyx welensii*, and *Operophtera brumata*, abiotic stress factors include salinity, drought, waterlogging, soil pollutants, cold, heat, carbon dioxide, ozone, and ultraviolet radiation. The review is structured following the Central Dogma of Molecular Biology and the omic cascade, from DNA (genomics, epigenomics, and DNA-based markers) to metabolites (metabolomics), through mRNA (transcriptomics) and proteins (proteomics). An integrated view of the different approaches, challenges, and future directions is critically discussed.

**Keywords:** *Quercus* spp.; biotic stress; abiotic stress; classical biochemistry; omics approaches; systems biology

## 1. Introduction

The genus *Quercus* (oak), family *Fagaceae*, comprises around 500 species, being one of the most important and dominant woody angiosperms in the Northern Hemisphere, with relevance from an environmental and economic point of view [1,2]. It includes shrubs and trees, either deciduous or evergreen, which are part of temperate and sub-tropical forests, sub-tropical and tropical savannah, and subtropical woodlands. They are also part of typical agrosilvopastoral systems, such as the Spanish "*dehesa*" and the Portuguese "*montado*", with *Quercus ilex* and *Quercus suber*, respectively, being the dominant species [3].

Forest trees are long-lived organisms that have played—since ancient times—an important role in the different human civilizations. They are considered key elements of the Earth's health and biodiversity, and are associated with our culture and spirituality, providing countless products and services. They must be conserved as unvaluable heritage for

future generations. However, humans are taking the opposite direction, with deforestation, clearing, or thinning of forests, which have been constant throughout history. According to Williams' estimation (reviewed in Seale [4]), humans have destroyed around 1.8 billion Ha over the past 5000 years, with an average of 360 Kha/year. The figures are more dramatic in modern times. Therefore, and according to Food and Agriculture Organization (FAO) [5], since 1990, 420 Mha of forest have been lost, with rates ranging from 16 Mha/year in the 1990s to 10 Mha in the last quinquennium.

Causes such as forest degradation, fragmentation, and loss, as well as anthropogenic ones (overexploitation, fires, and agricultural land conversions, among others), are the main reasons for the high increase in tree mortality and deforestation [6]. Moreover, adverse environmental factors play an important negative role in forest conservation. Such factors, which are termed stresses, are physical, chemical, or biological agents that impose restrictions for the growth and development, and depending on their intensity and duration, they may cause death [7]. Nowadays, the decline syndrome, affecting the genus *Quercus* and other species, is a good example, and an international concern. Oak decline is a complex syndrome in which several damaging biotic and abiotic agents interact, and together with improper practices, old individuals, and a lack of regeneration, it is causing progressive and massive dieback. The interrelation between abiotic (e.g., drought and high temperature) and biotic stresses (e.g., fungi and insects) is complex, and not properly explained from an experimental point of view. Oak decline diseases are examined in Europe, the Middle East, and North America [8]. Oak trees are predisposed to acute oak decline (AOD) attacks by bacterial species and environmental factors [9,10]. In England, AOD affects mature *Quercus robur* and *Quercus petraea* trees, triggering stem bleeds from vertical fissures on trunks and inner bark necrosis caused by gram-negative bacteria such as *Gibbsiella quercinecans* and *Brenneria goodwinii*, and, to a lesser extent, *Rahnella victoriana* and *Lonsdalea britannica* [11–14]. On the other hand, the *Phytophthora* spp. is an oomycete considered to be one of the most lethal pathogens described in oak trees [15,16]. In Spain and Portugal, the synergy between *Phytophthora cinnamomi* Rands and long drought periods is increasing *Q. ilex* and *Q. suber* mortality [17–19]. Other pathogens affecting oaks, such as *Phytium spiculum* Paul and *Biscogniauxia mediterranea* (De Not) Kuntze, are also considered to be part of the oak decline syndrome [20,21].

Extreme environmental conditions in a climate change scenario represent an issue of great relevance that will determine species survival and migration, as well as the forest composition, structure, and functionality [22,23]. Simulation models predict an increase in both temperature and the frequency of severe drought episodes [24,25], or in northern latitudes the increased wetness, which will determine forest vulnerability. How can we afford the conservation of forest and forest resources under the present and future environmental restrictions? There must be political, technical, and scientific solutions, all of which must be integrated, for obtaining sustainable management, conservation, and rational exploitation. In the 21st century, biotechnology is taking a pivotal role for facing this challenge and finding solutions. Biotechnology is defined as the use of techniques based on the knowledge of living beings, especially at the molecular level. There are different strategies, including classical genetic improvement programs, genetic engineering, searches for agrochemicals, and the exploitation of biodiversity. The latter strategy is considered to be the feasible alternative at short term in the current state of knowledge that can be used with *Quercus*, as the genus comprises non-domesticated and recalcitrance species with a long biological cycle. In this direction, a main objective is to characterize the phenotypic diversity of the species in terms of growth characteristics, productivity, and stress tolerance, among others, as a step prior to the selection of elite or plus genotypes that can be used in reforestation, conservation, and breeding programs.

This manuscript intends to be a review about the use of classical biochemistry and modern omics techniques (genomics, transcriptomics, proteomics, and metabolomics) employed so far to study the response of *Quercus* spp. to biotic and abiotic stresses (Table 1), and to understand how the genotype plus environment and epigenetics contribute to

tolerant or resistant phenotypes. The genetic and epigenetic factors of healthy or diseased individuals remain poorly investigated and are thus almost unknown in *Quercus* spp. The current selection of elite trees in reforestation, conservation, and breeding programs requires the use of molecular analysis related to growth and productivity, as well as to responses to environmental factors. Plants genetic variability can be analyzed following the biological information according to the Central Dogma of Molecular Biology and the omic cascade in eukaryotes, as can be observed in *Q. ilex* (Figure 1). Central Dogma was enunciated by Crick [26] and define the relation between DNA, RNA and proteins, but nowadays different disciplines are interpreted their own "Central Dogma" including the relation with the omics cascade [27–30] to the final output, the phenotype. This information can be determined at different hierarchical levels of organization from individuals to populations and species by using different approximations and parameters, such as classical biochemistry (isozymes and secondary metabolites), and molecular biology, including the modern omics techniques (genomics, transcriptomics, proteomics, and metabolomics), together with DNA-based markers and epigenetic techniques.

**Table 1.** Molecular stress studies carried out on the genus *Quercus* in the last 5 years. This table also includes those studies carried out at genomic and proteomic levels that are not under stress conditions to indicate the current status in this genus.

| | Habitus | Technique/Assay | Abiotic or Biotic Stress | Reference |
|---|---|---|---|---|
| **Classical Biochemistry** | | | | |
| *Q. brantii* | Deciduous | Pigments, proline, total phenolic and flavonoids content, lipid peroxidation, ROS determination, antioxidant enzymes and PAL activity | Drought | Jafarnia et al. [31] |
| *Q. cerris* | Deciduous | Pigments and proline content, lipid peroxidation | Drought and $O_3$ | Cotrozzi et al. [32] |
| | | Pigments and proline content, lipid peroxidation, sugar and ABA content | Drought and $O_3$ | Cotrozzi et al. [33] |
| *Q. coccifera* | Evergreen | Pigments content | Drought and cold | García-Plazaola et al. [34] |
| *Q. ilex* | Evergreen | Pigments and proline content, lipid peroxidation | Drought and $O_3$ | Cotrozzi et al. [32] |
| | | Pigments content | Drought and cold | García-Plazaola et al. [34] |
| | | Pigments content, ROS determination and antioxidant enzymes activity | Soil pollutants (Pb and Cd) | Arena et al. [35] |
| | | ROS, proline and phytohormones determination | Drought and $O_3$ | Cotrozzi et al. [36] |
| | | ABA determination | Drought | Peguero-Pina et al. [37] |
| | | Pigments, sugar, total phenolic and amino acid content | Drought | San-Eufrasio et al. [38] |
| *Q. infectoria* | Semi evergreen | Proline, sugar, total phenolic and flavonoids content, lipid peroxidation, ROS determination, antioxidant enzymes and PAL activity | Drought and charcoal disease | Ghanbary et al. [39] |
| *Q. libani* | Deciduous | Proline, sugar, total phenolic and flavonoids content, lipid peroxidation, ROS determination, antioxidant enzymes and PAL activity | Drought and charcoal disease | Ghanbary et al. [39] |
| *Q. pubescens* | Deciduous | Pigments and proline content, lipid peroxidation | Drought and $O_3$ | Cotrozzi et al. [32] |
| | | Pigments content | Drought and cold | García-Plazaola et al. [34] |

**Table 1.** *Cont.*

| | Habitus | Technique/Assay | Abiotic or Biotic Stress | Reference |
|---|---|---|---|---|
| **Classical Biochemistry** | | | | |
| *Q. petraea* | Deciduous | Lignin, sugar, cellulose, lipid peroxidation and amino acids content, ROS determination, and GR activity | Partial pressure pCO$_2$ | Arab et al. [40] |
| *Q. suber* | Evergreen | Starch, sugar, total phenolic content | Drought and UV radiation | Diaz-Guerra et al. [41] |
| **Genome** | | | | |
| *Q. aliena* | Deciduous | Chloroplast genome sequencing | - | Yang et al. [42] |
| *Q. aliena* var. *acuteserrata* | Deciduous | Chloroplast genome sequencing | - | Yang et al. [42] |
| *Q. baronii* | Evergreen | Chloroplast genome sequencing | - | Yang et al. [42] |
| *Q. bawanglingensis* | Evergreen | Chloroplast genome sequencing | - | Liu et al. [43] |
| *Q. dentata* | Deciduous | Chloroplast genome sequencing | - | Pang et al. [44] |
| *Q. dolicholepis* | Evergreen | Chloroplast genome sequencing | - | Yang et al. [42] |
| *Q. fabri* | Deciduous | Chloroplast genome sequencing | - | Pang et al. [44] |
| *Q. gambelii* | Deciduous | Chloroplast genome sequencing | - | Pang et al. [44] |
| *Q. glandulifera* var. *brevipetiolata* | Deciduous | Chloroplast genome sequencing | - | Pang et al. [44] |
| *Q. lobata* | Deciduous | Whole genome sequencing | - | Sork et al. [45] |
| *Q. macrocarpa* | Deciduous | Chloroplast genome sequencing | - | Pang et al. [44] |
| *Q. palustris* | Deciduous | Chloroplast genome sequencing | - | Pang et al. [44] |
| *Q. phillyraeoides* | Evergreen | Chloroplast genome sequencing | - | Pang et al. [44] |
| *Q. robur* | Deciduous | Whole genome sequencing | - | Plomion et al. [46,47] |
| *Q. rubra* | Deciduous | Chloroplast genome sequencing | - | Pang et al. [44] |
| *Q. serrata* | Deciduous | Chloroplast genome sequencing | - | Pang et al. [44] |
| *Q. stellata* | Deciduous | Chloroplast genome sequencing | - | Pang et al. [44] |
| *Q. suber* | Evergreen | Whole genome sequencing | - | Ramos et al. [48] |
| *Q. variabilis* | Deciduous | Mitochondrial and chloroplast genome sequencing | - | Yang et al. [42]; Bi et al. [49]; Pang et al. [44] |
| *Q. wutaishanica* | Deciduous | Chloroplast genome sequencing | - | Pang et al. [44] |
| **DNA Based Markers** | | | | |
| *Q. dentata* | Deciduous | SSR | Climate adaptation | Nagamitsu et al. [50] |
| *Q. ilex* | Evergreen | SSR | Climate adaptation | Fernandez i Marti et al. [51] |
| *Q. lobata* | Deciduous | SNP | Climate adaptation | Sork et al. [52]; Gugger et al. [53]; Browne et al. [54] |
| *Q. mongolica* | Deciduous | SSR | Climate adaptation | Nagamitsu et al. [50] |
| *Q. oleoides* | Evergreen | SNP | Cold | Meireles et al. [55] |
| | | SSR | Drought | Ramírez-Valiente et al. [56] |
| *Q. petraea* | Deciduous | SNP | Climate adaptation | Rellstab et al. [57] |
| | | SNP | Climate adaptation | Truffaut et al. [58] |
| *Q. robur* | Deciduous | SNP | Climate adaptation | Truffaut et al. [58] |
| *Q. rubra* | Deciduous | SSR | Drought | Lind-Riehl and Gailing, [59]; Khodwekar and Gailing [60] |
| *Q. suber* | Evergreen | SNP | Climate adaptation | Pina-Martins et al. [61] |

**Table 1.** *Cont.*

| | Habitus | Technique/Assay | Abiotic or Biotic Stress | Reference |
|---|---|---|---|---|
| **Epigenome** | | | | |
| *Q. lobata* | Deciduous | Single methylation variants | Climate gradients | Gugger et al. [53] |
| | | DNA Demethylation by 5-Azacytidine | Climate adaptation | Browne et al. [62] |
| *Q. suber* | Evergreen | Single methylation variants | Climate adaptation | Inácio et al. [63] |
| **Transcriptome** | | | | |
| *Q. berberidifolia* | Evergreen | RNA-Seq | Drought | Oney-Birol et al. [64] |
| *Q. cornelius-mulleri* | Evergreen | RNA-Seq | Drought | Oney-Birol et al. [64] |
| *Q. engelmannii,* | Semi evergreen | RNA-Seq | Drought | Oney-Birol et al. [64] |
| *Q. ilex* | Evergreen | RNA-Seq | - | Guerrero-Sánchez et al. [65,66] |
| | | RT-qPCR | *Phytophthora cinnamomi* | Gallardo et al. [67] |
| | | RNA-Seq | Salt and $O_3$ | Natali et al. [68] |
| | | RT-qPCR | Drought | Kotrade et al. [69] |
| | | RNA-Seq | Drought | Madritsch et al. [70] |
| *Q. lobata* | Deciduous | RNA-Seq | Drought | Gugger et al. [71]; Mead et al. [72] |
| *Q. pubescens* | Deciduous | RNA-Seq | Drought | Kotrade et al. [69]; Madritsch et al. [70] |
| *Q. petraea* | Deciduous | RNA-Seq | Waterlogging | Le Provost et al. [73] |
| *Q. robur* | Deciduous | RNA-Seq | Waterlogging | Le Provost et al. [73] |
| | | RT-qPCR | Drought | Kotrade et al. [69] |
| | | RNA-Seq | Drought | Madritsch et al. [70] |
| *Q. rubra* | Deciduous | RNA-Seq | $O_3$ | Soltani et al. [74] |
| *Q. suber* | Evergreen | RNA-Seq | Drought | Magalhães et al. [75] |
| **Proteome** | | | | |
| *Q. ilex* | Evergreen | 2-DE MALDI TOF/TOF | Drought | Simova-Stoilova et al. [76] |
| *Q. suber* | Evergreen | 2D-DIGE MALDI-TOF/TOF | Ectomycorrhizal | Sebastiana et al. [77] |
| **Metabolome** | | | | |
| *Q. alba* | Deciduous | HPLC-Orbitrap-MS | Drought | Suseela et al. [78] |
| *Q. ilex* | Evergreen | LC-Orbitrap-MS | Drought | Rivas-Ubach et al. [79]; Gargallo-Garriga et al. [80] |
| | | GC-MS | Drought | Mu et al. [81]; Rodríguez-Calcerrada et al. [82] |
| *Q. suber* | Evergreen | GC-MS | Drought | Haberstroh et al. [83] |
| | | LC-Orbitrap-MS | Drought | Almeida et al. [84] |
| | | GC-MS | *Cerambyx welensii* | Sánchez-Osorio et al. [85] |
| *Q. pubescens* | Deciduous | PTR-TOF-MS | Drought | Saunier et al. [86] |
| | | PTR-MS/ GC-MS | Drought | Genard-Zielinski et al. [87] |
| *Q. pyrenaica* | Deciduous | GC-MS | Drought and $CO_2$ | Aranda et al. [88] |
| *Q. rubra* | Deciduous | LC-MS | Drought | Top et al. [89] |
| | | HPLC-Orbitrap-MS | Drought | Suseela et al. [78] |
| *Q. robur* | Deciduous | TD-GC-MS | *Operophtera brumata* | Volf et al. [90] |

Abbreviations: Reactive oxygen species, ROS; phenylalanine ammonia lyase, PAL; abscisic acid, ABA; glutathione reductase, GR; Simple Sequence Repeat, SSR; single nucleotide polymorphism, SNP; RNA sequencing, RNA-Seq; Reverse Transcription-quantitative PCR, RT-qPCR; ozone, $O_3$; carbon dioxide, $CO_2$; 2-dimensional gel electrophoresis, 2-DE; Matrix-Assisted Laser Desorption/Ionization, MALDI; time-of-flight, TOF; Mass Spectrometry, MS; High-Performance Liquid Chromatography, HPLC; Proton Transfer Reaction, PTR; Gas Chromatography, GC; Thermal Desorption, TD; and Liquid Chromatograph, LC.

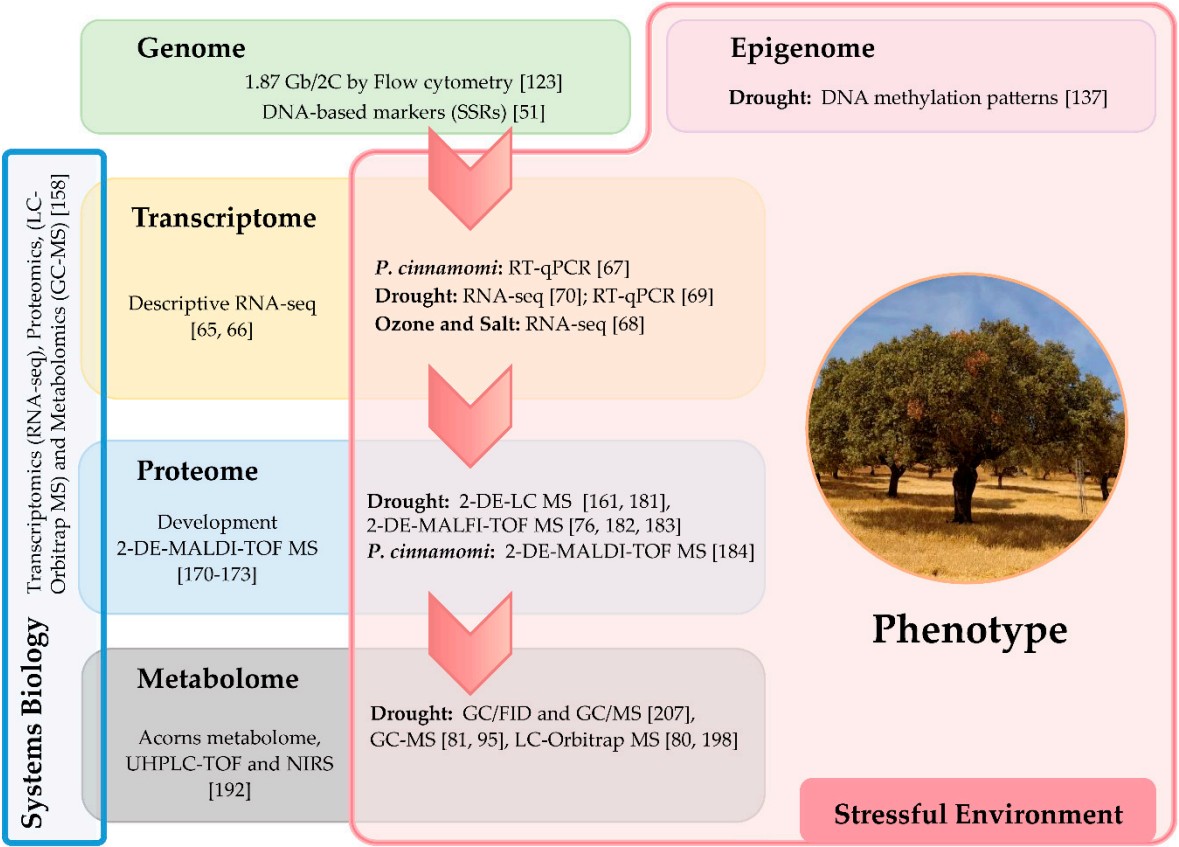

**Figure 1.** Central Dogma of Molecular Biology and omic cascade in *Quercus ilex* to characterize the responses to biotic and abiotic stresses using molecular approaches. The studies analyzed under stress in the different omics are located within the red "stressful environment" box. Abbreviations: Simple Sequence Repeats, SSRs; RNA sequencing, RNA-seq; Reverse Transcription-quantitative PCR, RT-qPCR; 2-dimensional gel electrophoresis, 2-DE; Matrix-Assisted Laser Desorption/Ionization, MALDI; time-of-flight, TOF; Mass Spectrometry, MS; Liquid Chromatography, LC; Ultra High-Performance Liquid Chromatography, UHPLC; near infrared spectroscopy, NIRS; Gas Chromatography, GC; and Flame Ionization Detector, FID.

## 2. Classical Biochemistry

A wide variety of biochemical studies have been conducted to study the mechanisms of the stress response in *Quercus* spp., with most of them being focused on abiotic stresses, especially drought. Stress affects many cellular processes, such as photosynthesis, signaling, primary and secondary metabolism, responses to oxidative damage, and osmoprotection.

Stressful conditions produce changes in the photosynthetic apparatus. The analysis of the pigment content, accompanied by photosynthetic measures, has been extensively researched in species of the genus *Quercus*. These analyses are useful for inferring possible degradation by oxidative damage (with chlorophylls being one of the targets), as well as photoprotection via xanthophyll cycle pigments. The photoprotective mechanisms involved in responses to abiotic stresses, such as drought and cold, have been reviewed for evergreen (e.g., *Q. suber*, *Quercus coccifera*, and *Q. ilex*) and deciduous (e.g., *Q. robur* and *Quercus pubescens*) *Quercus* spp. [34]. Both deciduous and evergreen *Quercus* species occurring in contrasting climates implement different photoprotective mechanisms under drought stress, in which energy dissipation via xanthophyll cycles plays an important role. Studies on *Q. pubescens* [91] and *Quercus brantii* [31] reported a decrease in the pigment content (Chla, Chlb, and carotenoids) suggesting impairment of the light-harvesting complex and a decline in the dissipation processes of thermal energy. Ramírez-Valiente et al. [92] studied the photoprotective mechanisms in four live oak species (*Quercus* series *Virentes*) in response to different climatic conditions (drought and low temperatures). By

studying pigment accumulation, the authors showed that *Quercus fusiformis* and *Quercus oleoides* increase the xanthophyll de-epoxidation state under water stress conditions when photoprotection is needed, while anthocyanins are used under cold stress in species with a limited freezing tolerance, such as *Quercus virginiana* and *Quercus geminate.* The performance of the photosynthetic machinery in response to an intense summer stress period was studied in three Mediterranean oaks (*Q. ilex*, *Q. coccifera*, and *Q. suber*). The photo-protection mechanisms were different in the three species. Only *Q. suber* exhibited a loss of chlorophyll, while the photosynthetic pigment composition remained unchanged in *Q. coccifera* and *Q. ilex*, despite interconversions within the xanthophyll cycle [93]. Similar results were reported by San-Eufrasio et al. [38] in *Q. ilex* under severe drought stress, with no significant changes in the photosynthetic pigment content suggesting little or no damage to the photosynthetic apparatus. Both Pb and Cd contamination have also been reported to increase the pigment content in *Q. ilex* to counteract the loss of photosynthetic surface area due to particle deposits in these contaminated environments [35].

The oxidative damage produced by stress has been studied in *Quercus* spp. by classical biochemical techniques such as reactive oxygen species (ROS) content analysis (e.g., $H_2O_2$ and $O_2^-$), lipid peroxidation (using the malondialdehyde (MDA) content), and the activity of antioxidant enzymes (superoxide dismutase, SOD; peroxidases, POD; catalase, CAT; ascorbate peroxidase, APX; glutathione reductase, GR; polyphenol oxidase activity, PPO; dehydroascorbate reductase, DR; and monodehydroascorbate reductase, MR). In *Quercus* spp., increased levels of ROS (mainly $O_2^\bullet$ or $H_2O_2$) have been observed in response to abiotic (drought, $O_3$, pollutants (Pb and Cd), and heat) and biotic (as charcoal disease caused by the fungal pathogens *Biscogniauxia mediterranea* and *Obolarina persica)* stresses [31,33,39,91,94], being more intense when these stresses are combined (drought and charcoal disease) [39,91]. However, a reduction in ROS in stresses with enhanced ROS scavenging such as $pCO_2$ has also been observed [40]. ROS cause the autocatalytic peroxidation of membrane lipids, which can be determined by the MDA content. Increased levels of superoxide radicals, hydrogen peroxide, and MDA were observed in several *Quercus* species in response to drought [31], drought and heat [91], drought and $O_3$ [32,33], and drought and charcoal disease (which causes the drying of parts of the canopy and ultimately leads to local or systemic necrosis of the tree) [39]. The antioxidant enzymes SOD and POD increased levels in *Q. brantii* in response to drought [31], *Q. pubescens* in response to drought and heat [91], and *Quercus infectoria* and *Quercus libani* in response to drought and charcoal disease [39]. Under severe drought stress, *Q. ilex* showed increased levels of APX, GR, and CAT [31,95], with the latter also being increased in response to pollutants [35].

The content of sugars and amino acids, in primary metabolism, and the phenolic content, as well as the key enzyme activity involved in this metabolic pathway, in secondary metabolism, have been widely analysed in this genus. Under drought conditions, the content of sugars and amino acids (mainly proline) was increased in *Q. brantii*, improving the osmoprotection and preventing water loss [31]. The combination of this abiotic stress with charcoal disease in *Q. libani* and *Q. infectoria,* or $O_3$ in *Quercus cerris*, also increased the content of these biomolecules [39]. Nevertheless, the content of sugars and amino acids was not altered in *Q. ilex* and *Q. suber* [38]. Secondary metabolism is normally stimulated under stress in the genus *Quercus*, with an increase in the total phenolic and flavonoid content or phenylalanine ammonia lyase (PAL) activity under drought or both drought and charcoal disease [31,38,39,95]. However, *Q. suber* did not show an increase of these compounds under drought or UV conditions [41].

A signaling process analysis by phytohormones has also been performed in this genus. An increase in the content of abscisic acid (ABA), considered a key hormone that mediates plant responses to adverse environmental stimuli, has been described for the genus *Quercus* under drought conditions, causing stomatal closure or premature leaf senescence [36,37,96]. The content of ABA, salicylic acid (SA), jasmonic acid (JA), and ethylene (ET) was determined in *Q. ilex* in response to drought and $O_3$, with all of them

reaching high concentrations at the initial stage of $O_3$ treatment, demonstrating a spatial and functional correlation between these compounds [36]. However, the JA levels reached higher values when drought and $O_3$ treatments were combined.

Many biochemical studies have been conducted on *Quercus* in response to abiotic stress, most of them aimed at studying changes in the photosynthetic apparatus and oxidative damage on leaves. Global biochemical analysis has been used to make inferences on biological aspects, but it is currently complemented by the application of specific techniques for the identification of compounds, as well as the massive analysis of proteins and metabolites. Studies that use these techniques beyond the global vision have been detailed in the omics layers in this review.

### 3. DNA-Based Markers

Following the Central Dogma of Molecular Biology, the first molecular layer is the DNA. DNA-based markers are widely used to estimate genetic diversity and germplasm identification. Initially, isozyme markers contributed to tree breeding programs [97]. However, currently, the use of DNA-based markers has achieved a high impact in forest tree programs. These markers can be grouped into hybridization-based DNA markers (RFLP), PCR-based DNA (e.g., sequence-characterized amplified regions (SCAR), amplified fragment length polymorphism (AFLP), random amplified polimorphic DNA (RAPD), simple-sequence repeats (SRR), and inter simple sequence repeats (ISSRs)), and DNA chip and sequencing-based DNA markers (e.g., single nucleotide polymorphism (SNP)) [98]. To characterize the response to stresses in the genus *Quercus*, investigations have mainly been performed with SSRs and SNPs in leaf tissue (reviewed in Müller and Gailing [99]). The drought tolerance has been widely studied with SSR markers in the genus *Quercus* [56,59,60,100]. Nagamitsu et al. [50] used nuclear SSRs in *Quercus dentata* and *Quercus mongolica* var. *crispula* to determine their adaptability to different environmental conditions in two different ecotypes, either oak coastal or inland forests in northern Japan, reporting a positive correlation between these DNA markers and environmental conditions. For *Q. ilex*, there is a study in which different Andalusian regions were determined by using a set of SSR markers (both nuclear and chloroplast) [51]. The eastern *Q. ilex* populations showed a higher level of drought tolerance than the western ones [101]. For *Quercus aquifolioides,* patterns of 381 SNPs from 65 candidate genes were analysed in relation to adaptation to climate change were analyzed [102]. Out of these SNPs, twelve and two genes were shown to be associated with drought and temperature, respectively. Candidate genes associated with environmental adaptation have also been reported in *Q. suber*, *Quercus lobata, Q. oleoides, Q. petraea,* and *Q. robur* by using SNPs [52,54,55,57,58,61,103]. For example, SNP 158 and SNP 168 located on a trehalase and peroxidase, respectively, play a role in drought stress [61]. Further, SNP markers have been used to position and localize putative candidate genes for drought tolerance on two genetic linkage maps of *Q. robur* and *Q. petraea*, with these being the first mapped genes on linkage maps of *Quercus* spp. [104].

Additionally, ISSR markers were used to determine the level of tolerance to oak dieback phenomena in *Q. brantii* [105]. These DNA-based markers provided useful information in the screening of oak seedlings to be used correctly in the restoration of damaged stands. A total of seven ISSR markers were associated with dieback phenomena. Metal toxicity, which is considered to be a major abiotic stressor, has been analysed in two-generation *Quercus rubra* populations using the RAPD marker system [106]. It was determined that the amount of bioavailable metal did not modify the genetic variation in *Q. rubra*, suggesting that two generations cannot be enough to cause change in the total DNA. The identification of quantitative trait loci (QTL) for tolerance-related traits is a useful tool for a future search of putative candidate genes for biotic and abiotic tolerance [107–109]. In *Q. robur* clones, five regions containing QTL specific to hypoxia responses were defined, which suggested waterlogging responses from a genetic point of view [108].

The potential of DNA-based markers is far from being exploited in the genus *Quercus*. So far, the majority of studies developed in these forest species have been related to drought;

however, the large battery of SSRs and SNPs already described as well as new batteries of these markers might give relevant information in response to other stresses [110,111].

## 4. Genomics

The genome is the complete set of genetic information within a single cell of an organism. Genomics focuses on the study and cataloguing of all the genes that an organism possesses, as well as their organization, structure, and function. Research on forest tree genomics has fallen behind that of model and crop systems. However, genomic research on forest trees has progressed significantly with the advent of next-generation sequencing technologies (NGS). Until the sequencing of the genome of black cottonwood (*Populus trichocarpa*), which was selected for its small genome size, no reference genome sequence was available in forest species [112]. Thereafter, the list of forest tree genomes has been increased with the sequencing of *Picea abies* [113], *Picea glauca* [114], *Eucalyptus grandis* [115], and *Pinus tadea* [116], producing more than 50 genomes of tree species that are available in the NCBI Genome database [117,118]. In the genus *Quercus*, three species have been sequenced (*Q. robur* [46,47], *Q. lobata* [45], and *Q. suber* [48]). The oak genome size was initially estimated by Feulgen cytophotometry [119] and then, by flow cytometry [120–122]. Focusing only on those oak species already sequenced, *Q. robur* and *Q. suber* displayed a 2C nuclear DNA of 1.88 and 1.91 pg with a 38.9 and 39.7 GC% content, respectively. Information on the genome of *Q. lobata* was published at later [45], being currently the version 3.0 of valley oak genome available at https://valleyoak.ucla.edu/genomic-resources/ (accessed on 1 March 2021). In addition, all *Quercus* spp. are formed of 12 pairs of chromosomes (2n = 2x = 24). By using NGS, the genome size of *Q. robur, Q. lobata,* and *Q. suber* was determined to be 1.45 Gb/2C, 1.15 Gb/2C, and 1.90 Gb/2C, respectively [45–48]. Progress on the *Q. ilex* genome is ongoing, datasets have been generated by the single-molecule real-time (SMRT) technology called PacBio. The genome datasets have been submitted to NCBI's BioProject repository (ID: 687489), but they are not yet public. The estimated genome size of *Q. ilex*, by flow cytometry, was approximately 1860 Mb/2C with a total length of 1.87 Gb/2C and, as expected, it is formed by 12 pairs of chromosomes [120,123–125].

The relevance of forest species genome sequencing is related to the huge genetic diversity observed to identify key genes and alleles controlling a large number of important traits. Much effort has been devoted to elucidating genomic backgrounds to enhance the response to biotic and abiotic stresses. In Gene Ontology (GO) term enrichment analysis in *Q. robur*, gene products involved in "*response to biotic stimulus* (GO:0009607)" and "*defense response* (GO:0006952)" were included in the best supported GO terms [47].

The chloroplast and mitochondrial genomes have been sequenced and annotated in the genus *Quercus* [42–44,49,126–128]. The whole mitochondrial genome has been only sequenced in *Quercus variabilis* [49]. The chloroplast genome has been described in several *Quercus* species (Table 1). For example, a study published in 2016 compared the chloroplast genomes of five *Quercus* spp. from China (*Quercus baronii*, *Quercus dolicholepis*, *Q. variabilis*, *Quercus aliena*, and *Q. aliena* var. *acuteserrata*) [42]. After analyzing nucleotide substitutions, indels, and repeats, a total of 19 variable regions were identified that could provide plastid markers for taxonomic, phylogenetic, and stress studies within this genus [42].

The availability of the sequenced genomes facilities the access to essential information about genes and gene products with their functions, transcript levels, cis-acting regulatory elements and alternative splicing patterns [129], as well as characterize novel phenotypes using a combination of whole-genome resequencing, linkage maps and microarrays shedding some light to the gene expression changes and introducing new SNPs related to interest traits [130].

## 5. Epigenomics

Epigenetics is the study of all processes (heritable or not) that modify gene expression or activity of transposable elements without modifying the DNA sequence [131]. Epigenetics is characterized as a fundamental mechanism that contributes to phenotypic variation

(gene regulation, development, the stress response, phenotypic diversity, and evolution). Epigenetic modifications (DNA methylation, histone modifications, and microRNA regulation) play an important role in the expression of stress-induced genes, contributing to a plant's potential to adapt to stresses [132,133]. DNA methylation has been widely assessed in plants by the silencing of gene expression [134]. Cytosine methylation, common form of post-replicative DNA modification seen in both bacteria and eukaryotes, occurs at CG, CHG and CHH sites, where H can be A, T, or C [134,135]. Moreover, other epigenetic mechanisms are considered in other layers of the central dogma such as histone modification in proteomics and RNA methylation in transcriptomics [29], although they have been less studied. Epigenetic variation contributes the phenotypic plasticity, adaptative capacity, and ability to persist in variable environments, which could be relevant for long-lived organisms with complex life cycles, such as forest trees. The epigenetic variation analysis of trees has gained much interest in recent years, recently reviewed in Amaral et al. [131]; however, few studies have investigated epigenetic variations related to climate adaptation in forest trees, in general, and in the genus *Quercus*, in particular. A high-quality genome sequence assembly in the sequencing of forest tree genomes may contribute to an increase in forest epigenetic studies. To date, the effect of biotic and abiotic stress at an epigenetic level has been studied in few *Quercus* spp., so the epigenetic changes that may occur under stress conditions remain largely unknown in this genus. Correia et al. [136] stated that epigenetic mechanisms such as DNA methylation and histone H3 acetylation are essential for the acclimation and survival of 8-month-old *Q. suber* plants at high temperature. In *Q. ilex*, Rico et al. [137] reported changes in the DNA methylation patterns in adult individuals exposed to drought for 12 years, indicating that these changes are associated with drought stress acclimation. DNA methylation variations associated with climate gradients have also been described in *Q. lobata* mature leaves and seedlings [53,62,138]. Browne et al. [62] reported a reduced in the genome-wide methylation induced by 5-Azacytidine in 5-month-old seedlings of the tree, *Q. lobata*, suggesting that removal of DNA methylation could affect plant adaptation to environmental change. As a first approach to inherited epigenetic analysis in *Quercus*, we have evaluated the global DNA methylation patterns in two individuals of *Q. ilex* located in Aldea de Cuenca (Córdoba, Andalusia, Spain) using the methylation sensitive amplification polymorphism (MSAP) technique, which is one of the most used methods for assessing DNA methylation changes in plants [139]. Different methylation patterns have preliminary been found between individuals and developmental stages (adult trees and seedlings). Our current results have also unveiled specific DNA methylation differences that could correspond to relatively stable changes and be involved in the development of epigenetic memory of the stress.

Recently, the existence of epigenetic priming, or individual stress memory triggering natural plant defenses [140,141], has been described in forest trees [133,142]. Initially, this term was associated with biotic stress (e.g., immunity against pathogens), but this phenomenon has also now been related to abiotic stress [143,144]. Most studies carried out to determine the epigenetic priming have been developed for the genera *Pinus*, *Picea*, and *Populus* [142,145–147]. For the genus *Quercus*, there is a lack of priming studies focused on biotic and abiotic stress from an epigenetic point of view. Recently, studies based on applying chemical elicitors, such as benzothiadiazole (BTH), have been applied to *Q. ilex* seeds and seedlings to induce epigenetic memory of defense responses (priming) that may increase resistance to abiotic (drought) or biotic (*P. cinnamomi*) future stress, but the results have not yet been published.

Currently, the interest by the tree epigenome is increasing with an emergent need to explore epigenetic diversity in response to stresses [131]. To date, the methodology applied to the genus *Quercus* provides overall information about global DNA methylation patterns associated to high temperature and drought. However, it is required a step forward in the epigenome knowledge with the identification of genes encoding epigenetic regulators (e.g., DNA methylases) and stress-induced changes in the expression of these genes to explain the contribution of epigenetic modifications to tree plasticity [148].

## 6. Transcriptomics

The second molecular layer within the Central Dogma of Molecular Biology in eukaryotes is the RNA. The transcriptome is the set of all RNA molecules, including messenger RNA (mRNA), ribosomal RNA (rRNA), transfer RNA (tRNA), small nuclear RNAs (snRNAs), small nucleolar RNAs (snoRNAs), and other noncoding RNA (ncRNA), produced in one cell or a population of cells [149]. The transcriptome, unlike the genome, is modifiable under different conditions (such as development stage, tissue, and environmental changes), being a promising layer for exploring the stress response. The techniques used to determine the gene expression are grouped into targeted (reverse transcription-quantitative PCR (RT-qPCR) and microarray) and untargeted (RNA sequencing (RNA-seq)) transcriptomic studies.

A targeted transcriptomic study is a useful tool for studying processes of interest, being of great utility, even after the arrival of NGS technologies. In *Quercus* spp., targeted transcriptomics studies started with RT-qPCR or microarrays (as used by Spieß et al. [150] in *Q. robur* under drought). There are recent studies of RT-qPCR in important processes such as cork formation in *Q. suber* under drought (e.g., the study of gene *QsMYB1* by Almeida et al. [151]), and the different expression of aquaporins in short-term waterlogging in *Q. petrea* and *Q. robur* (tolerant) [152] or under biotic stress produced by *P. cinnamomi* in the expression of defense genes in *Q. suber* [153] or tannin synthesis genes in *Q. ilex* [67]. In these studies, it is important to highlight the analysis of the alternative splicing produced by stress, showing how allelic gene variants play a fundamental role in the response to stress. Almeida et al. [151] observed the preference for the QsMYB1.1 variant upon heat stress, while in drought, QsMYB1.2 was the predominant one. In the same sense, Gallardo et al. [67] observed how variant 2 of shikimate dehydrogenase (SDH2) was more active under mechanical defoliation.

NGS was developed to study the genome, allowing the analysis of RNA through the sequencing of complementary DNA (cDNA) [154]. This non-targeted method, called RNA-Seq, has clear advantages over previous approaches (such as microarrays) and has revolutionized our understanding of the complex and dynamic nature of the transcriptome [149]. For these reasons, this review is focused on them. For the genus *Quercus*, in recent years, thanks to the affordability of these analyses, numerous studies of non-targeted transcriptomic analysis have been conducted. RNA-seq analyses of the genus *Quercus* were initiated in 2010 in *Q. petrea* and Q. *robur* [155], using buds, leaves, and wood. In these analyses, a high enrichment in the molecular function "*response to stress*" was observed, being the second one with the highest number of transcripts (10.8%), showing a similar percentage to that in the analysis of the *Q. robur* DF159 clone [156]. The *Q. ilex* transcriptome [65,66] (analyzed in a mix of embryos, leaves, and roots) also exhibited an enrichment in "*response to stress*", being the gene ontology category with the highest number of transcripts (5405 transcripts) in the Biological Processes. However, a transcriptomic analysis of more than 21 types of tissue (covering genes expressed in multiple tissues, developmental stages, and stress conditions) carried out in *Q. suber* showed a low enrichment of this biological process, being below 3% of those transcripts in "*response to stress*" [157]. These descriptive transcriptomic studies are of great help for the identification and creation of specific protein databases for *Quercus* non-sequenced species [158,159], achieving better protein identification than using generalist databases.

Comparative non-targeted transcriptomics analysis under stress allows the global transcriptomics response to be deepened, with numerous examples within the genus *Quercus*. Le-provost et al. [73] studied in seedlings the response to flooding in *Q. petraea* and *Q. robur*, and showed that *Q. robur* was more tolerant due to the activation of suberin biosynthetic, ABA, and ethylene pathways related to the formation of adaptive structures (hypertrophied lenticels and adventitious roots), and the rapid upregulation of the fermentation pathway, which helps to maintain energy production and overcome hypoxia in this tolerant species [73]. Under drought, a non-targeted analysis on leaf of different *Q. lobata* populations demonstrated how seedlings from different locations respond to different

environmental conditions [71,72]. Under drought, the populations overexpressed the same pathways (e.g., abiotic stress or senescence processes) [71], although the authors observed how some gene expression, such as heat shock protein (HSP) 2-like chaperones (related to memory to stress) or metabolism-related proteins (e.g., phosphoglycerate kinase), were population-dependent. In the same line, Mead et al. [72] observed differences between populations adapted to warmer climatic conditions under drought, where gene expression for ribosomal proteins and for chloroplast and mitochondrial structures were differentially increased, while gene expression for DNA replication were differentially decreased between populations. This may be due to the fact that the response pathways are different among populations adapted to combined stress than those that are usually only under one stress (drought) [72]. Another application in transcriptional study is to analyze adaptive introgression using genes associated with environmental factors. In this sense, Oney-Birol et al. [64] analyzed the hybridization and introgression in adult trees of *Quercus engelmannii*, *Quercus berberidifolia*, and *Quercus cornelius-mulleri* using allelic variables in drought genes. Out of 139 drought candidate genes, eleven were selected as possible drought-associated species-specific adaptation genes, such as cellulose synthase and HSP 70kDa protein 10. Within European oaks, the transcriptomic response in drought has also been studied in the seedling roots of *Q. suber* [75] and seedling leafs of *Q. robur*, *Q. pubescences*, and *Q. ilex* (from less to more tolerant) [70]. Magalhães et al. [75] observed how the activation of the signaling response mediated by ABA- and ABF-dependent genes was of key importance under drought for *Q. suber*. Furthermore, Madritsch et al. [70] confirmed in 9 year-old plants that, under drought, there are species-specific responses to drought. In *Q. robur,* the important pathways were the ROS scavenging machinery, ABA-mediated gene expression, and cellular osmotic adjustment, as well as the initiation of the catabolic pathway, possibly due to high stress with the onset of senescence. In *Q. pubescens,* processes such as ROS scavenging or minor catabolic processes with a decrease in photosynthesis were detected, but heat-shock proteins and lignification pathways were expressed. The more tolerant *Q. ilex* also presented overexpression of the ROS pathway, although it differentially overexpressed genes relating to a new cell wall remodeling pathway [70]. The exposure to $O_3$ (alone or in combination with other stresses) has also been reported in the *Quercus* spp. Soltani et al. [74] analyzed the exposure to three doses of $O_3$ (150 ppb, 225 ppb, and 300 ppb) in *Q. rubra* seedlings. Exposure to the highest concentrations resulted in the activation of defense gene expression by altering the metabolic pathways of carbohydrates, amino acids, lipids, and terpenoids (such as the preferred biosynthesis for mono-, di-, and tetraterpenes under stress), as well as in modifying key processes, such as photosynthesis and the ATP production pathway. Furthermore, oxidative stress caused by $O_3$ produced an increase in ROS which the plant faces with the increase of detoxification pathways, as seen in the increase in glutathione pathway overexpression [74]. In *Q. ilex* seedlings, Natali et al. [68] studied the combined effect of salt and $O_3$, as well as in individual cases, showing unique responses in terms of the expression in single or combined stress situations with treatment-specific expressed transcripts (such as sucrose-phosphate synthase only being overexpressed in combined stress). The transcriptomic data showed common stress molecular responses (such as ROS scavenging or signaling by phytohormones), but treatment-specific differences in expression in key pathways, such as photosynthesis, cell wall remodeling, and sugar and lipid metabolism. In conclusion, the impact of saline stress was more severe than with $O_3$ alone, but in combination, it had a strong effect on gene overexpression and suppression. Finally, from the RNA-seq analysis, it is possible to observe the selection of genes that do not vary their expression throughout the stress. These genes, considered as candidate reference genes, could improve normalization of RT-qPCR over classical reference genes. The comparative transcriptome analysis obtained among *Q. robur*, *Q. pubescences*, and *Q. ilex* [70] selected specific candidates (such as serine/threonine-protein phosphatase PP1 for *Q. ilex*) to be used for reference genes for gene expression normalization in a RT-qPCR analysis of each *Quercus* [69].

The affordability of RNA-seq has allowed a significant advance in the analysis of stress in the *Quercus* genus, being predominant the studies in drought. These results have usually been obtained in seedlings, with few examples in adult trees, but of great importance for the advancement of research. Moreover, the analysis of the splicing variables that occur under different stresses is a field that is beginning to be studied in this genus, where the progress to global non-targeted analysis is positioned as a promising field of study.

## 7. Proteomics

The transcriptome is translated into functional proteins that ultimately determine what happens in a living system. Proteomics intends to study the whole proteome or the sum of all proteins from an organism, tissue, and cell. Proteins undergo different changes (post-translational modifications, proteolysis, recycling, or multi-complex formation), which are key to the regulation of cellular processes [160]. As environmentally-sensitive layers, proteomic analysis is a powerful tool for the discovery of the key molecular mechanisms involved in stress. Proteomics has been scarcely used for research on forest species and even less so in the genus *Quercus*. Within this genus, to the best of our knowledge, holm oak (*Q. ilex*) is the most studied at the proteomic level and, to a lesser extent, other species, such as *Q. suber* and *Q. rubra*. Proteomics studies have been carried out in *Q. ilex* with a methodological optimization purpose [161–165], proteotyping or proteome descriptive characterization [158,166–169], and seed maturation and germination [170–173]. Proteomics studies in *Q. suber* have mainly addressed the cork quality [174,175] and somatic embryogenesis [176,177]. In addition, proteomics is emerging as a promising tool for identifying allergens in the pollen of *Quercus* spp. [178,179]. Most proteomic studies on the responses to stresses have been performed in *Q. ilex*, most being limited to the use of the gel-based (one or two-dimensional gel electrophoresis) coupled to MALDI-TOF MS (Matrix-Assisted Laser Desorption/Ionization time-of-flight mass spectrometry), and to a lesser extent, shotgun gel-free (LC-MSMS) platforms. All of the proteomics studies performed in *Q. ilex* have been reviewed in Rey et al. [123,180], with many research in abiotic and biotic stresses related to decline syndromes (drought and *P. cinnamomi* infestation). The first proteomics works in *Q. ilex* were focused on the variation of the leaf proteome between provenances in response to drought [161]. Further studies were addressed to characterize and select those genotypes with high levels of tolerance to drought [76,181–183] and resistance to *P. cinnamomi* [184]. Overall, differences in tolerance were related in *Q. ilex* at the provenance level, and a general tendency to decrease proteins of photosynthesis and ATP synthesis upon water withholding was observed [182]. In response to *P. cinnamomi*, two *Q. ilex* provenances were studied [184]. Physiological and proteomic analyses revealed differential responses of the provenances studied, with a tendency to decrease proteins of photosynthesis, amino acid metabolism, and stress-related proteins, mainly in the most susceptible one. However, some proteins related to starch biosynthesis, glycolysis, and stress-related peroxiredoxin were increased upon inoculation. The authors concluded that holm oak's response to *P. cinnamomi* resembles the drought-stress response [184].

Besides the aforementioned works carried out in *Q. ilex*, only one proteomic work has been performed for *Q. robur* to study the effect of one season of drought stress, simulating the conditions of a dry summer for young oak trees [185]. Phenotypical and proteomics (Two-Dimensional Difference Gel Electrophoresis, 2D-DIGE) results revealed that plants were initially able to cope with the stress, but prolonged drought periods produced an accumulation of osmoactive compounds (such as mono- and di-saccharides) in the foliar tissue, adjusting their metabolism for some time, with the cost of reducing growth and productivity. It is also worth mentioning in this section the importance of ectomycorrhizal symbiosis in forest species. The protein profile of *Q. suber* in response to ectomycorrhizae formation was studied using proteomics and biochemical approaches [77]. The comparative analysis of mycorrhizal and nonmycorrhizal plants revealed no differences at the foliar level. However, an increased allocation of carbohydrates from the plant to the fungus to sustain the symbiosis was observed in roots. In addition, protein unfolding, the attenuation

of defense reactions, increased nutrient mobilization from the plant–fungus interface (N and P), cytoskeleton rearrangements, and induction of plant cell wall loosening for fungal root accommodation in colonized roots were also suggested. These results corroborate the potential of mycorrhizal inoculation to improve the cork oak forest resistance capacity to cope with future climate change [77].

As a future prospect, proteomics has great potential in plant biological research in general, and in the genus *Quercus* in particular, which has been little exploited thus far. Most of proteomics studies have been carried out in *Q. ilex*, with stress studies being mostly limited to leaves using gel-based approach. Through the use of state-of-the-art proteomics techniques, together with a deep analysis conducted through subcellular fractionation, the study of post-translational modifications (PTMs), and target proteomic techniques (selected/multiple-reaction monitoring (SRM/MRM), parallel-reaction monitoring (PRM) or sequential window acquisition of all theoretical mass spectra (SWATH)), a range of possibilities will emerge that will allow for deeper knowledge on the molecular mechanisms underlying defense responses in the genus *Quercus* against biotic and abiotic stresses.

## 8. Metabolomics

Metabolomics can be considered as the omics layer the closest to the phenotype, being the last layer within the omic cascade. Metabolomics is also one of the most environmentally-sensitive layers, which allows a snapshot of any stress situation to be obtained [186–189]. Therefore, in the renewed search for biomarkers for stress tolerance, it is one of the most promising omics for the discovery of candidates [190,191] and useful for the identification of nutraceutical and bioactive acorn compounds [192], as possible food products for our diet in this genus. Metabolomics stress research was started with a classical biochemical analysis of total compounds within the different plant organisms, as has already been described above. These global biochemical analyses were completed with a targeted metabolomic analysis focused on the identification of compounds of interest [193].

The metabolomic analysis of phenolic compounds is a relevant field within the genus *Quercus*. These compounds have multiple protective functions against environmental stress [89], being important in the synthesis of the cell wall under temperature stress [194]. In recent years, within the phenolic compounds, target studies aimed at analyzing the production of tannins (condensed and hydrolysable) in different stress conditions. By using LC-MS, Top et al. [89] determined the molecular composition of condensed and hydrolysable tannins produced under a high temperature and drought in *Q. rubra*. In this study, plants grown under a single stress (high temperature or drought) produced more tannins per unit of leaf mass than those grown under favorable conditions, being higher for the combination of these stresses. Additionally, within this targeted study of the metabolome, volatile organic compounds (VOCs) emitted by oaks were also analyzed by using GS-MS. In trees, the emission of isoprenoids is widespread, with the genus *Quercus* being unique in evolved species emitting isoprene, monoterpene, or non-isoprenoids [195]. Oaks from the western hemisphere (*Quercus alba*, *Q. lobata*, and *Q. rubra*) and two from the eastern hemisphere (*Q. robur*, *Q. pubescens*) mainly emit isoprene, whereas some oaks from Mediterranean environments (*Q. ilex*, *Q. coccifera*, and *Q. suber*) emit monoterpenes [195,196]. The emission of isoprenoids helps protect against oxidative stress in plants, as protection against high temperatures [197] or in interaction with other organisms for different reasons (herbivory, pollination, or communication with other plants) [198]. The study of these emissions ranges from the investigation of seasonal emissions variations [87,199–201] to the change in emissions under stress conditions [202]. The emission of isoprenoids under drought stress within the genus *Quercus* is controversial [87,95]. In most studies reported, emissions under a slight stress were increased, but were reduced under extreme drought [87,203]. Despite this, studies developed in adults trees of *Q. pubescens* have identified an increase in isoprene emissions in situations of mild water stress [86,200], as well as a reduction in them [204,205]. Furthermore, conditions of a high temperature have a stimulating effect on the release of carbon from isoprene synthesis and mitochondrial respiration. However,

this effect was attenuated under drought [82]. In adult trees of *Q. ilex,* the emission of terpenes has mainly been analyzed under drought using GS-MS [81,95]. The temperature is an incentive for the emission of terpenes in this species, although it is very sensitive to drought, only displaying an increase of particular terpene emissions at moderate stress, and a slight reduction in the total amount of terpenes emitted [95]. The increase occurred for the main terpenes, such as limonene [81,95] and $\alpha$-pinene [81]. In prolonged drought stress, *Q. suber* adult trees reduced the total terpenoid emission, remaining increased in some specific terpenoids, such as $\gamma$-terpinene [83]. This may be due to its concrete importance in the adaptation to stress [83,206]. In this context, *Q. suber* trees under biotic stress by *Cerambyx welensii* modified the monoterpene emission profile, emitting more limonene as opposed to the non-stressed profile dominated by the pinene type [85]. The emissions in *Q. suber* do not greatly depend on leaf water potential or air temperature [83] as in *Q. ilex* [207]. A comparison of emissions in drought versus flooding has also been studied in *Q. robur* [208]. According to the authors, the total VOC emissions were decreased in drought and increased in flooding, although the isoprene emissions were not altered in both stresses. The VOC emission induction against herbivory has recently been reviewed for trees [209]. For example, Volf et al. [90] showed an increase in the emission of 45 VOCs, mainly terpenes (4 mono- and 21-sesquiterpenes), which produce a decrease in caterpillars' appetite for methyl jasmonate-treated branches, as well as a possible attraction of predators in *Q. robur*.

In recent years, the advent of massive data analysis has allowed metabolomic analysis from a global perspective, which has permitted a more in-depth understanding of the metabolome in the presence of different stresses. Among the platforms used, LC-MS coupled to an LTQ Orbitrap XL is one of the most frequently used, permitting the analysis of a high number of compounds. Rivas-Ubach et al. [198] determined an increase in the flavonoid concentration under drought, as well as in the total content of soluble sugars, in *Q. ilex* adult trees. The accumulation of these compounds in drought may be due to their role as antioxidants (flavonoids) and in osmotic regulation (soluble sugars), making leaves more attractive to folivores [198]. Additionally, small-sized *Q. ilex* trees showed a greater carbohydrate accumulation and terpene concentration than larger trees in drought [79]. The response to drought in *Q. ilex* seedlings has also been studied at the level of root exudates, due to its impact on the plant, other neighboring plants, and soil properties, being mainly identified compounds of primary and secondary metabolism [80]. The root metabolism under drought was analyzed in *Q. rubra* and *Q. alba* seedlings and it was reported that the amount of lignins was not modified in the fine roots, although it did increase in others part of the plants as in leaves [78]. Therefore, adaptation to drought was facilitated in part by changes in the content and composition of tannins [78]. In *Q. rubra* trees, Susavella et al. [210] also analyzed the change in the metabolic production and resorption in leaves against single (high temperature or drought) or combined stresses. The metabolomic profile of the single stresses was not comparable to the combined one, showing an increase of the amino acid content in both, but with different key amino acids per treatment [210] (as spartic acid, methionine and lysine under combined stress). The combined stress of moderate drought and $CO_2$ was studied in *Quercus pyrenaica* seedlings by Aranda et al. [88], who only observed significant differences in recovery. This study differs from those performed in *Q. rubra*, where the concentration of amino acids increased in drought [210]. This denotes the possible species-specific response within the genus *Quercus.*

VOCs emission is well studied in the genus under different stresses in adult trees, predominantly drought. However, in non-targeted metabolomics (more recently used), the majority of studies are in seedlings. The studies show a great variability in the emission or accumulation of metabolites within the genus under stress, showing the existence of key compounds specific for species and stress. This validates the metabolome as one of the most promising molecular layers in the search for markers of tolerance to different stresses in the *Quercus* genre.

## 9. Integration and Systems Biology

The current trend in molecular analysis is to go one step beyond the mere analysis of single and independent omics layers using a Systems Biology approach. System biology is the study of biological systems whose behavior cannot be reduced to the linear sum of the functions of their parts [211]. For this purpose, computational modelling of molecular systems and an integrative interpretation of the different omics layers (such as transcriptomics, proteomics or metabolomics) are used [212]. This approach can potentially provide a more complete picture of the stress response in plants by exposing the interconnection between the different molecular levels and provided a global vision of the stress. Moreover, it is possible to improve the robustness of the molecular analysis obtained under stress. The Systems Biology approach has been carried out through a global integration of the different omics layers, as well as with classical biochemistry, physiological, or phenotypic data. This approach has been little studied in the genus *Quercus*, as in general in trees, although a few studies have already been performed in other forest species, such as *Pinus* spp. [191,213–215]. The first contact with this approach in the genus *Quercus* was the integration of different omics layers (transcriptomics, proteomics, and metabolomics) for the most complete network reconstruction of metabolic pathways in *Q. ilex* spp. [158]. The *Q. ilex* metabolic network included carbohydrate and energy metabolism, amino acid metabolism, lipid metabolism, nucleotide metabolism, and the biosynthesis of secondary metabolites, with the tricarboxylic acid cycle being the pathway most represented. Under stress, Almeida et al. [84] analyzed the response to drought in *Q. suber* seedlings by means of an integrated analysis of the metabolome, together with physiological parameters, showing different metabolomic profiles under drought and post rehydration. This study highlights the fundamental role of secondary metabolism in the different processes (in short- and long-term drought stress and recovery), showing specific metabolites for each moment, and proposing key metabolites in the different parts of the processes.

## 10. Conclusions and Future Perspectives

The present review is, to our knowledge, together with Kremer et al. [216], the only manuscripts that deal with molecular studies developed in the genus *Quercus*. This kind of studies started with the development of an EST catalogue performed in *Q. petraea* and *Q. robur* to be used in ecosystem genomics studies twenty years ago [155]. It is remarkable that the first whole genome sequencing of a species included in the genus *Quercus*, *Q. robur*, has been carried out five year ago [46], in comparison to the first genome sequencing in plants, *A. thaliana* carried out 20 years ago in 2000 [217], or in forest trees, *Populus trichocarpa* in 2006 [112]. To date, massive analysis methodologies have been developed (genomics, epigenomics, transcriptomics, proteomics, and metabolomic) implemented with other information (such as classical biochemistry or physiological data) that have helped to understand key concepts such as stress in the biology of forest species [129,218].

The integration of research molecular techniques in the study of *Quercus* spp. will support the most advances in the direction of Systems Biology contributing to our molecular stress understanding as has been performed in other forest species such as poplar or pine [213,214,219]. The available information on this genus is disperse and fragmented (as in general for the majority of forest species [218]), so that coordinated and cooperative multidisciplinary research consortiums, including ecologists, ecophysiologists, geneticists, forestry engineers, and molecular biologist, should be pursued in the future to understand, monitor, and predict functional genetic diversity. The molecular interpretation in the genus *Quercus*, as in other non-model species, has been challenging due to the absence of reference genotypes. The high molecular variability observed as well as other methodological aspects that complicate experimentation (such as the complexity in obtaining clones, seed recalcitrance or its long life cycle [220]) have contributed to the lack of molecular information in this genus. So, there is an imperative need to build consensus reference populations for each species in all countries [221], as well as establishing defined phenotypes for comparison and integration of data.

Nevertheless, the molecular techniques used are starting to give answers to processes related to development, germination, and genes or gene products related to environmental resilience and productivity, as well as elucidating biosynthetic pathways and regulatory networks. In comparison to the development of molecular approaches in model plant species such as *A. thaliana*, *Oryza sativa* or *P. trichocarpa*, the genus *Quercus* is still in its infancy. *Quercus* stress studies have focused on simple experimental systems in seedlings analyzing the response in a single organ (leaves) under controlled conditions. This allows us to establish partial and fixed frames, from which it is difficult to extrapolate to natural environments. So, the combination of controlled conditions with outdoor experiments across environmental gradients (e.g., provenances of different environments) will give us a more accurate view of the responses [218] triggered in the face of multiple stresses. Multiple stresses providing a more real understanding of how they occur in nature (as occurs for oak decline syndrome) and how responses differ from those observed by the same stresses separately.

The availability of a higher number of sequenced genomes will allow characterizing and cataloguing the natural biodiversity, as well as developing molecular markers based on DNA and epigenetic marks to establish correlations between variability and the responses to stresses and resilience. In the most environmentally sensitive omics layers (transcriptomics, proteomics, and metabolomics), key putative genes can be identified, and individual genes can be functionally validated by using model organisms, such as *Arabidopsis*. Therefore, the use all the described molecular approaches will facilitate the identification of those elite genotypes that, after clonal propagation, can be used in reforestation, conservation, and improvement programs as well as to mitigate the effect of climate change.

**Author Contributions:** Conceptualization, M.E. and J.V.J.-N.; writing—original draft preparation, M.E., M.Á.C. and M.-D.R.; writing—review and editing, M.E., M.Á.C., J.V.J.-N. and M.-D.R.; supervision, J.V.J.-N. All authors have read and agreed to the published version of the manuscript.

**Funding:** This work was supported by grant ENCINOMICS-2 PID2019-10908RB-100 from the Spanish Ministry of Science, Innovation and Universities. M.E., M.Á.C., and M.-D.R. are grateful for awards of Juan de la Cierva-Formación (FJCI-2017-31613), Ramón y Cajal (RYC-2017-23706), and Juan de la Cierva-Incorporación (IJC2018-035272-I) contracts by the Spanish Ministry of Science, Innovation and Universities, respectively.

**Institutional Review Board Statement:** Not applicable.

**Conflicts of Interest:** The authors declare no conflict of interest.

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
