# Peer review of "Molecular Research on Stress Responses in Quercus spp.: From Classical Biochemistry to Systems Biology through Omics Analysis"

_forests, doi:10.3390/f12030364_

Round 1

Reviewer 1 Report

The manuscript by Escandón et al makes a review of the molecular research in oaks, mainly regarding stress responses. This review include biochemistry, genomic, transcriptomic, proteomic and metabolomic studies, and gives an integrated view of the molecular research in the genus Quercus. The theme fits the scope of Forests journal, and I would recommend some revisions.

My main concern is about the use of the analogy of the Central Dogma of Biology. Metabolites and epigenetics hardly fit the Central Dogma. Retroelements, for instance are against this Dogma since the information flows in inverse order. Therefore, rephrasing of all the sentences invoking the Central Dogma is needed.

Also, there are some misperception in the text because stress responses are in the title as the focus of the review, but in some cases, there are also other examples. This should be elucidated, and a less strength should be made on the stress responses, for instance in the title.

Furthermore, the order of the sections should be altered. After the introduction the review should follow with genomics, transcriptomics, proteomics and then biochemistry and metabolomics, finalizing with systems biology.

In detail:

Line 70-71 “….caused by Gibbsiella quercinecans and Brenneria goodwinii, and, to a lesser extent, Rahnella 70 victoriana and Lonsdalea Britannica…” indicate what organisms are these species, and use lowercase in britannica.

Line 71-72 “Oak trees…..” Move this phrase to line 68 before “In England ….”

Line 77 Remove the comma after syndrome

Line 90  “.... is considered the only alternative….” Reformulate this, because it is not the only alternative. It is, perhaps, the most important in the short term, since is already available  and is, therefore, more feasible.

Figure 1 is a little bit confusing. In genome, the reference 115 is not appropriate for the flow cytometry and genome size since this is a review paper on proteomics.

Page 3- The title of the table is not correct since there are much more works on oaks in the last 5 years. If an emphasis wants to be made in stress conditions the title must be rephrased, or the other references should be included.

Page 5- Reference 56. This kind of references i.e. posters or conference participations that are not publicly available should be avoided because readers cannot access them.

In the epigenome section this reference should be added:

Inácio, V., Barros, P.M., Costa, A., Roussado, C., Gonçalves, E., Costa, R., Graça, J., Oliveira, M.M. and Morais-Cecílio, L., 2017. Differential DNA methylation patterns are related to phellogen origin and quality of Quercus suber cork. PloS one, 12(1), p.e0169018.

Line 166. The charcoal disease should be explained, and the causal agent should be indicated

Line 166-167 the sentence “Although, observing also ….” seems incomplete

Line 171-172 “The antioxidant enzymes SOD and POD 171 were activated ….” These enzymes are not present in the absence of drought? please reformulate the sentence

Line 182-183 “Nevertheless, the content of these biomol- 182 ecules was not altered in Q. ilex and Q. suber” Under which conditions? Clarify

Line 209 “(e.g. RADseq, SNP)”. In fact, RADseq is a technique to obtain SNPs. All the sequencing base methods will detect SNPs, i.e., SNPs arise from very different ways, and it should not be mistaken the polymorphism (SNP), and the way it is obtained (in this case RADseq). Reformulate

Line 212 “…it have been…” Correct to it has been

Line 225,226 All these SNP (158, 168, 1589 and 233) are from ref 55 and not from 99 and 100. The last two SNPs are associated with temperature (seasonality and annual mean) and not directly to stress. 158 and 168 are associated with mean temperature of the driest quarter and to the precipitation of the driest quarter, and not with drought (at least the first one). Reformulate the sentence.

Line 228 “SNP markers has been also….” Correct to SNP markers have been also

Line 235,236 “These markers identified from 1 (ISSR165) to 17 alleles (ISSR016) with a 235 fragment size within the range 250-1400 bp.” This is not relevant. Please delete this sentence.

Line 254 It should also be referred the Picea abies genome which was the first of the gymnosperms.

Nystedt, B., Street, N., Wetterbom, A. et al. The Norway spruce genome sequence and conifer genome evolution. Nature 497, 579–584 (2013). https://doi.org/10.1038/nature1221

Line 255 “…being a total of 52 genomes of tree species…” Because new genomes are being released every day this kind of exact numbers should be avoided. It can be reformulated to more than 50 genomes ….

Line 257,258 “Firstly, oak genome size was estimated by flow cytometry [113,114].” Indeed, the first calculation of Q. robur genome was by Favre JM, Brown S.  1996.  A flow cytometric evaluation of the nuclear DNA content and GC percent in genomes of European oak species.  Annales des Sciences Forestieres 53: 915-917.

Also, th first study on oaks genome sizes was from Olszewska MJ, Osiecka R. 1984 Relationship between 2C DNA content, systematic position and level of  DNA endoreplication during differentiation of root parenchyma in dicot shrubs and trees-comparison with herbaceous sp. Biochem Physiol  Pflanzen 179: 641-657 on Quercus sessilis, although the technique used was Feulgen cytophotometry. Please reformulate the references.

Line 266,267 “… BioProject repository (ID: 687489),…” This reference cannot be found while searching bioprojects. This should be removed or corrected.

Line 267, 268  “Rey et al., [115] determined the genome size of Q. ilex by flow cytometry…” This reference is not suitable for this data since it is a review paper in proteomics. You should refer

Zoldoš V, Papeš D, Brown SC, Panaud O, Šiljak-Yakovlev S.  1998. Genome size and base composition of seven Quercus species: inter- and intra-population variation.  Genome 41: 162-168. since they were the first to calculate the genome size of this species.

Also, the chromosomes of Quercus ilex were first reported by Zoldos, V., Papes, D., Cerbah, M., Panaud, O., Besendorfer, V. and Siljak-Yakovlev, S., 1999. Molecular-cytogenetic studies of ribosomal genes and heterochromatin reveal conserved genome organization among 11 Quercus species. Theoretical and Applied Genetics, 99(6), pp.969-977. and then by Ribeiro, T., Loureiro, J., Santos, C. and Morais-Cecílio, L., 2011. Evolution of rDNA FISH patterns in the Fagaceae. Tree genetics & genomes, 7(6), pp.1113-1122. You should include these references

Line 269-270 “The relevance of the forest species genome sequencing is related to the huge genetic diversity observed in them”. Explain better

Line 276-279. These data are irrelevant. Make a more general statement about chloroplast and mitochondrial genomes

Page 10 Epigenomics. Since cork quality has been mentioned in several transcriptomic and proteomic studies it also should be mentioned here in the epigenomic studies

Inácio, V., Barros, P.M., Costa, A., Roussado, C., Gonçalves, E., Costa, R., Graça, J., Oliveira, M.M. and Morais-Cecílio, L., 2017. Differential DNA methylation patterns are related to phellogen origin and quality of Quercus suber cork. PloS one, 12(1), p.e0169018.

Line 288 “Epigenetic modifications (DNA methylation, histone modifications and small RNAs)…” Although small RNAs are involved in the epigenetic regulation they cannot be considered an epigenetic modification. This sentence should be reformulated.

Line 301 “…. for the establishment of Q. suber at high temperature.” This study is on acclimation and survival, and don’t involve the process of establishment since all the research was made under laboratory conditions. Reformulate.

Line 305-312 Please see above the comment about reference 56

Line 345 The sentence seems to lack something

Line 351 “… RNA-Seq sequencing,…” remove sequencing as RNA-seq stands for RNA sequencing.

Line 368 “…. using generalist bases.” Databases?

Line 379 “…some genes…” Perhaps gene expression

Line 383,384 genes should be replaced by reads or gene expression

Line 405-411 This paragraph should be moved to the end of the section. Chang housekeeping genes to reference genes for gene expression normalization.

Line 413 “(150 ppb, 225 ppb and 300pb)” 300 ppb?

Line 511-512 “… emitted by the genus Quercus….” It is not the genus that emit VOCs,  but each species. Therefore, change to oaks: “emitted by oaks”

Line 524-515 “….Oaks from the western hemisphere (Q. robur, Q. pubescens, Q. alba, Q. lobata, Q. rubra). Q. roburand Q. pubescens are not from the Western hemisphere, since this hemisphere corresponds of the regions located west to the Greenwich meridian. Reformulate

Line 597 and 600  “ This research will…) Which research? Please clarify

Line 602 “Q. Cornelius-mulleri …” lowercase in Cornelius

Line 621 “ A previously stated,…” As previously stated

“…tree species are the most recalcitrant …” Comparing with what? Clarify

Line 623 “… epigenetics can be determined….” This is not correct. What is determination of epigenetics? Reformulate

Line 627 “… such as Arabidopsis.” And also Populus, since it is also a model organism with biological processes much closer to other tree species than the annual herbaceous Arabidopsis.

Line 631 – 633 Reformulate the last paragraph since it lacks something.

Author Response

Reviewer #1

Comments and Suggestions for Authors

The manuscript by Escandón et al makes a review of the molecular research in oaks, mainly regarding stress responses. This review includes biochemistry, genomic, transcriptomic, proteomic and metabolomic studies, and gives an integrated view of the molecular research in the genus Quercus. The theme fits the scope of Forests journal, and I would recommend some revisions.

Firstly, we really appreciate all the helpful comments and suggestions of the three reviewers. In addition, we apologize to the reviewers because we think that they have not reviewed the English edited version. All the answers have been included in bold.

My main concern is about the use of the analogy of the Central Dogma of Biology. Metabolites and epigenetics hardly fit the Central Dogma. Retroelements, for instance are against this Dogma since the information flows in inverse order. Therefore, rephrasing of all the sentences invoking the Central Dogma is needed.

Answer: We present a modern version of the central dogma in which reversed transfer of genetic information, from RNA to DNA should be considered. It is applicable to retrovirus, but not to eukaryotes, plants included. The metabolome, as part of the proteome (i.e. structural proteins), can be considered within the phenotype. Epigenetics is one of the mechanisms (ncRNAs, DNA and histone modification) regulating gene expression. It is possible that we have used a free interpretation of the central dogma together with the omic cascade (based in articles as Di Leo et al., 2007; Patti et al., 2012; Fu et al., 2014; or Srivastava, 2019), trying to justify why the different -omics approaches, as the information flux in plants. According to the referee comments we have rephrased sentences referring to the central dogma.

Di Leo A, Claudino W, Colangiuli D, Bessi S, Pestrin M, Biganzoli L. New strategies to identify molecular markers predicting chemotherapy activity and toxicity in breast cancer. Ann Oncol. Suppl 12:xii8-14 (2007)

Patti G, Yanes O. & Siuzdak G. Metabolomics: the apogee of the omics trilogy. Nat Rev Mol Cell Biol 13, 263–269 (2012)

Fu Y, Dominissini D, Rechavi G. et al. Gene expression regulation mediated through reversible m6A RNA methylation. Nat Rev Genet 15, 293–306 (2014)

Srivastava S. Emerging Insights into the Metabolic Alterations in Aging Using Metabolomics. Metabolites, 9(12): 301 (2019)

Also, there are some misperceptions in the text because stress responses are in the title as the focus of the review, but in some cases, there are also other examples. This should be elucidated, and a less strength should be made on the stress responses, for instance in the title.

Answer: The review is focused on stress in the genus Quercus, although for an exhaustive analysis we have included other works that are not under stress to validate the molecular approach (such as genome sequencing), or to have a more accurate perception of the state of the art of research in the different species of the Quercus genus. This is also specified in legend of Table 1.  

Furthermore, the order of the sections should be altered. After the introduction the review should follow with genomics, transcriptomics, proteomics and then biochemistry and metabolomics, finalizing with systems biology.

Answer: It could be another possibility for organizing the review, but we decided to follow this organization due to a historical reason as techniques and studies have appeared, from classical biochemistry to -omics approaches. In addition, classical biochemistry analysis includes varied analyses at the molecular level that could fit into different omics layers.

In detail:

Line 70-71 “….caused by Gibbsiella quercinecans and Brenneria goodwinii, and, to a lesser extent, Rahnella victoriana and Lonsdalea Britannica…” indicate what organisms are these species, and use lowercase in britannica.

Answer: We have included that they are gram-negative bacteria (lines 73).

Line 71-72 “Oak trees…..” Move this phrase to line 68 before “In England ….”

Answer: That phrase has been moved as you propose.

Line 77 Remove the comma after syndrome

Answer: It has been removed.

Line 90  “.... is considered the only alternative….” Reformulate this, because it is not the only alternative. It is, perhaps, the most important in the short term, since is already available and is, therefore, more feasible.

Answer: Reformulate to “The latter strategy is considered to be the feasible alternative at short term in the current state of knowledge that can be used with Quercus

Figure 1 is a little bit confusing. In genome, the reference 115 is not appropriate for the flow cytometry and genome size since this is a review paper on proteomics.

Answer: We agree with the reviewer that the reference 115 is a review mainly focused on proteomics. However, in Rey et al., 2020, we included new data that were not published previously as the size of holm oak genome size by flow cytometry.

Page 3- The title of the table is not correct since there are much more works on oaks in the last 5 years. If an emphasis wants to be made in stress conditions the title must be rephrased, or the other references should be included.

Answer: The title of the table was modified to “Table 1. Molecular stress studies carried out on the genus Quercus in the last 5 years. This table also includes those studies carried out at genomic and proteomic levels that are not under stress conditions to indicate the current status in this genus”.

Page 5- Reference 56. This kind of references i.e. posters or conference participations that are not publicly available should be avoided because readers cannot access them.

Answer: Following the comments of R1 and considering that this reference is a contribution to a conference, we have removed it in the current version of the review.

In the epigenome section this reference should be added:

Inácio, V., Barros, P.M., Costa, A., Roussado, C., Gonçalves, E., Costa, R., Graça, J., Oliveira, M.M. and Morais-Cecílio, L., 2017. Differential DNA methylation patterns are related to phellogen origin and quality of Quercus suber cork. PloS one, 12(1), p.e0169018.

Answer: We did not consider including this reference because there is not a direct relationship between Quercus-Epigenetics-Stress. In this paper, authors propose different cytosine methylation patterns related to cork quality avoiding any developmental and/or environmentally related variation in DNA methylation. However, we have included it as reviewer indicated.

Line 166. The charcoal disease should be explained, and the causal agent should be indicated

Answer: The causal agents (Biscogniauxia mediterranea and Obolarina persica) were included in line 176, and the effect in line 183-184.

Line 166-167 the sentence “Although, observing also ….” seems incomplete

Answer: We have rephrased the sentence to “However, a reduction in ROS in stresses with enhanced ROS scavenging such as pCO2 has also been observed.” The sentence is in green in the track manuscript as it has already been modified in the revised English version.

Line 171-172 “The antioxidant enzymes SOD and POD 171 were activated ….” These enzymes are not present in the absence of drought? please reformulate the sentence

Answer: They are complex as present in multiple isoforms. We have modified to “The antioxidant enzymes SOD and POD increased levels in Q. brantii in response to drought”.

Line 182-183 “Nevertheless, the content of these biomolecules was not altered in Q. ilex and Q. suber” Under which conditions? Clarify

Answer: We have clarified the meaning of “these biomolecules”. We meant the content of sugars and amino acids rather than the content of these biomolecules.

Line 209 “(e.g. RADseq, SNP)”. In fact, RADseq is a technique to obtain SNPs. All the sequencing base methods will detect SNPs, i.e., SNPs arise from very different ways, and it should not be mistaken the polymorphism (SNP), and the way it is obtained (in this case RADseq). Reformulate

Answer: This has been reformulated considering that RADseq is a methodology to identify SNPs rathe than a kind of DNA markers.

Line 212 “…it have been…” Correct to it has been

Answer: Done.

Line 225,226 All these SNP (158, 168, 1589 and 233) are from ref 55 and not from 99 and 100. The last two SNPs are associated with temperature (seasonality and annual mean) and not directly to stress. 158 and 168 are associated with mean temperature of the driest quarter and to the precipitation of the driest quarter, and not with drought (at least the first one). Reformulate the sentence.

Answer: We have modified the references 99 and 100 by the reference 55. Considering that 1589 and 233 are not directly related to stress, we have removed the whole sentence.

Line 228 “SNP markers has been also….” Correct to SNP markers have been also.

Answer: Done.

Line 235,236 “These markers identified from 1 (ISSR165) to 17 alleles (ISSR016) with a 235-fragment size within the range 250-1400 bp.” This is not relevant. Please delete this sentence.

Answer: Done.

Line 254 It should also be referred the Picea abies genome which was the first of the gymnosperms.

Nystedt, B., Street, N., Wetterbom, A. et al. The Norway spruce genome sequence and conifer genome evolution. Nature 497, 579–584 (2013). https://doi.org/10.1038/nature1221

Answer: This reference has been included.

Line 255 “…being a total of 52 genomes of tree species…” Because new genomes are being released every day this kind of exact numbers should be avoided. It can be reformulated to more than 50 genomes ….

Answer: Thank you for this comment. It has been modified in the main text.

Line 257,258 “Firstly, oak genome size was estimated by flow cytometry [113,114].” Indeed, the first calculation of Q. robur genome was by Favre JM, Brown S.  1996.  A flow cytometric evaluation of the nuclear DNA content and GC percent in genomes of European oak species.  Annales des Sciences Forestieres 53: 915-917.

Answer: Thank you for this comment. It has been included in the review.

Also, the first study on oaks genome sizes was from Olszewska MJ, Osiecka R. 1984 Relationship between 2C DNA content, systematic position and level of DNA endoreplication during differentiation of root parenchyma in dicot shrubs and trees-comparison with herbaceous sp. Biochem Physiol  Pflanzen 179: 641-657 on Quercus sessilis, although the technique used was Feulgen cytophotometry. Please reformulate the references.

Answer: Thank you so much for clarifying that the first study on oak genome sizes was reported in Olszewska and Osiecka (1984).

Line 266,267 “… BioProject repository (ID: 687489),…” This reference cannot be found while searching bioprojects. This should be removed or corrected.

Answer: As we say in the main text, raw data have been deposited in a public database, but they are not available yet because we are working on the bioinformatic part. We considered that this information could be relevant to the scientific community to know that the Q. ilex genome is coming soon. However, if the reviewer 1 considers that this should be not included in this review, we will remove it.

Line 267, 268  “Rey et al., [115] determined the genome size of Q. ilex by flow cytometry…” This reference is not suitable for this data since it is a review paper in proteomics. You should refer

Zoldoš V, Papeš D, Brown SC, Panaud O, Šiljak-Yakovlev S.  1998. Genome size and base composition of seven Quercus species: inter- and intra-population variation.  Genome 41: 162-168. since they were the first to calculate the genome size of this species.

Answer: As we mentioned above, in Rey et al., 2020, we included new data that were not published previously as the size of holm oak genome size by flow cytometry and the morphology of holm oak chromosomes. Apart from this, we agree with R1 and we have also included Zoldoš et al., 1998 in the review.

Also, the chromosomes of Quercus ilex were first reported by Zoldos, V., Papes, D., Cerbah, M., Panaud, O., Besendorfer, V. and Siljak-Yakovlev, S., 1999. Molecular-cytogenetic studies of ribosomal genes and heterochromatin reveal conserved genome organization among 11 Quercus species. Theoretical and Applied Genetics, 99(6), pp.969-977. and then by Ribeiro, T., Loureiro, J., Santos, C. and Morais-Cecílio, L., 2011. Evolution of rDNA FISH patterns in the Fagaceae. Tree genetics & genomes, 7(6), pp.1113-1122. You should include these references.

Answer: we have rephrased the idea about genome size and chromosomes of Q. ilex to include the following references: Zoldoš et al., 1998; Zoldoš et al., 1999 and Ribeiro et al., 2011.

Line 269-270 “The relevance of the forest species genome sequencing is related to the huge genetic diversity observed in them”. Explain better.

Answer: We have clarified this sentence. “The relevance of forest species genome sequencing is related to the huge genetic diversity observed to identify key genes and alleles controlling a large number of important traits”.

Line 276-279. These data are irrelevant. Make a more general statement about chloroplast and mitochondrial genomes.

Answer: Due to the presence of irrelevant data, we have modified this paragraph to simply it.

Page 10 Epigenomics. Since cork quality has been mentioned in several transcriptomic and proteomic studies it also should be mentioned here in the epigenomic studies

Inácio, V., Barros, P.M., Costa, A., Roussado, C., Gonçalves, E., Costa, R., Graça, J., Oliveira, M.M. and Morais-Cecílio, L., 2017. Differential DNA methylation patterns are related to phellogen origin and quality of Quercus suber cork. PloS one, 12(1), p.e0169018.

Answer: This reference has been included in the review. See answer included above.

Line 288 “Epigenetic modifications (DNA methylation, histone modifications and small RNAs)…” Although small RNAs are involved in the epigenetic regulation they cannot be considered an epigenetic modification. This sentence should be reformulated.

Answer: We considered microRNAs as epigenetic modifications, as previously was reported in Morales et al., 2017 Biomolecular concepts 8: 20-212, Sato et al., 2011 The FEBS journal 278:1598-1609, among others.

Morales S, Monzo M, & Navarro A. Epigenetic regulation mechanisms of microRNA expression. Biomolecular concepts8(5-6), 203-212. Sato, F., Tsuchiya, S., Meltzer, S. J., & Shimizu, K. (2011). MicroRNAs and epigenetics. The FEBS journal278(10), 1598-1609 (2017)

Line 301 “…. for the establishment of Q. suber at high temperature.” This study is on acclimation and survival, and don’t involve the process of establishment since all the research was made under laboratory conditions. Reformulate.

Answer: Thank you for your suggestions. We have clarified it in the main text.

Line 305-312 Please see above the comment about reference 56

Answer: The reference 56 was deleted.

Line 345 The sentence seems to lack something

Answer: Modified in the revised English version to “Almeida et al. (2020) observed the preference for the QsMYB1.1 variant upon heat stress, while in drought, QsMYB1.2 was the predominant one”. The sentence is in green in the track manuscript.

Line 351 “… RNA-Seq sequencing,…” remove sequencing as RNA-seq stands for RNA sequencing.

Answer: Done.

Line 368 “…. using generalist bases.” Databases?

Answer: Modified to “generalist databases”.

Line 379 “…some genes…” Perhaps gene expression

Answer: Modified to “gene expression”.

Line 383,384 genes should be replaced by reads or gene expression

Answer: Modified to “gene expression”.

Line 405-411 This paragraph should be moved to the end of the section. Chang housekeeping genes to reference genes for gene expression normalization.

Answer: Modified to “Finally, from the RNA-seq analysis, it is possible to observe the selection of genes that do not vary their expression throughout the stress. These genes, considered as candidate reference genes, could improve normalization of RT-qPCR over classical reference genes. The comparative transcriptome analysis obtained among Q. robur, Q. pubescences, and Q. ilex selected specific candidates (such as serine/threonine-protein phosphatase PP1 for Q. ilex) to be used for reference genes for gene expression normalization in a RT-qPCR analy-sis of each Quercus.” and moved.

Line 413 “(150 ppb, 225 ppb and 300pb)” 300 ppb?

Answer: Modified to “300 ppb”.

Line 511-512 “… emitted by the genus Quercus….” It is not the genus that emit VOCs,  but each species. Therefore, change to oaks: “emitted by oaks”

Answer: Modified to “emitted by oaks”.

Line 524-515 “….Oaks from the western hemisphere (Q. robur, Q. pubescens, Q. alba, Q. lobata, Q. rubra). Q. robu rand Q. pubescens are not from the Western hemisphere, since this hemisphere corresponds of the regions located west to the Greenwich meridian. Reformulate

Answer:  Modified to “Oaks from the western hemisphere (Q. robur, Q. pubescens, Quercus alba, Q. lobata, and Q. rubra) and two from the eastern hemisphere (Q. robur, Q. pubescens) mainly emit isoprene, whereas some oaks from Mediterranean environments (Q. ilex, Q. coccifera and Q. suber) emit monoterpenes.”

Line 597 and 600  “ This research will…) Which research? Please clarify

Answer: Modified to “the molecular research will open new …”

Line 602 “Q. Cornelius-mulleri …” lowercase in Cornelius

Answer: Done.

Line 621 “ A previously stated,…” As previously stated

Answer: Done.

“…tree species are the most recalcitrant …” Comparing with what? Clarify

Answer: The conclusion was rewritten following the reviewers' recommendations, and this sentence was eliminated.

Line 623 “… epigenetics can be determined….” This is not correct. What is determination of epigenetics? Reformulate

Answer: Modified to “epigenetics can be elucidated”

Line 627 “… such as Arabidopsis.” And also Populus, since it is also a model organism with biological processes much closer to other tree species than the annual herbaceous Arabidopsis.

Answer: Thank you for your suggestions. Populus has been added.

Line 631 – 633 Reformulate the last paragraph since it lacks something.

Answer: The final conclusion was rewritten following the reviewers' recommendations.

Reviewer 2 Report

This manuscript provides an excellent and thorough summary of the literature on stress response in oaks.  Researchers interested in oaks will find it very useful. The main sections of the paper correspond to organizational levels and each one is very thorough.  Typically, reviews will try to provide sub-conclusions within each section, and this paper does not, which is a minor limitation of the review.  

The authors use a few major terms without defining them.  The terms are:

1) Central Dogma of Molecular Biology--I did not know there was a central dogma.  Please define the central dogma and cite a reference.  Without the definition, it is not clear how use of the term adds to the organization of the paper.

2) Epigenetic priming--another term that the authors don't define. 

3). Systems Biology.  How do the authors define this term?  Cite a reference.  Be more specific on what this approach will reveal about stress response.  

4) I find the word "omics" to be jargon.  The title does not need it. In fact the title could be simplified to "Molecular and genomic research on stress response in the tree genus Quercus". 

The section on epigenetics might benefit by some definitions on epigenetics from the broader literature. The authors might consider using background for 1 or 2 of these papers (but not all):

  • Bewick, A. J., Ji, L., Niederhuth, C. E., Willing, E.-M., Hofmeister, B. T., Shi, X., . . . Schmitz, R. J. (2016). On the origin and evolutionary consequences of gene body DNA methylation. Proceedings of the National Academy of Sciences, 113(32), 9111-9116. doi:10.1073/pnas.1604666113
  • Law, J. A., & Jacobsen, S. E. (2009). Dynamic DNA Methylation. Science, 323(5921), 1568-1569.
  • Bewick, A. J., & Schmitz, R. J. (2017). Gene body DNA methylation in plants. Current Opinion in Plant Biology, 36, 103-110. doi:10.1016/j.pbi.2016.12.007
  • Johannes, F., & Schmitz, R. J. (2019). Spontaneous epimutations in plants. New Phytologist, 221(3), 1253-1259. doi:10.1111/nph.15434
  • Niederhuth, C. E., Bewick, A. J., Ji, L., Alabady, M. S., Do Kim, K., Li, Q., . . . Udall, J. A. (2016). Genome Biology, 17(1), 194.
  • Niederhuth, C. E., & Schmitz, R. J. (2014). Molecular Plant, 7(3), 472-480. doi:DOI 10.1093/mp/sst165

Conclusions and Future Perspectives: 

I find this section to be the weakest part of the paper.  Because the paper is a summary of various kinds of research rather than an exploration of specific questions or hypotheses, it is difficult for the authors to draw conclusions.  This section could be shortened to one paragraph with a few conclusions, which the authors would need to develop, and a second paragraph on future perspectives.  The authors are vague on this as well so I encourage them to think about what the questions are and what the approach should be.  I do not find their current ideas compelling.  

Minor comments:

The paper has scattered typos throughout the manuscript and inconsistent use of italics for species names in the text and especially literature cited.  I do not list fully, but below are a few I caught as examples. They typically would escape spell check functions.

Line 199: change "this" to "these"

Line 278:  change "on" to "one"

One more paper that the authors might include in the epigenetic section is: 

Browne, L., Mead, A., Horn, C., Chang, K., A. Celikkol, Z., L. Henriquez, C., . . . Sork, V. L. (2020). Experimental DNA demethylation associates with changes in growth and gene expression of oak tree seedlings. G3: Genes|Genomes|Genetics, 10(3), 1019-1028. doi:10.1534/g3.119.400770

In Table 1, the authors cite two papers on Q. lobata for climate adaptation.  They may also want to include: 

Browne, L., Wright, J. W., Fitz-Gibbon, S., Gugger, P. F., & Sork, V. L. (2019). Adaptational lag to temperature in valley oak (Quercus lobata) can be mitigated by genome-informed assisted gene flow. Proceedings of the National Academy of Sciences, 116(50), 25179-25185. doi:10.1073/pnas.1908771116

In Table 1, the authors list valley oak genome V. 1 (Sork et al. 2016). The valley oak genome 3.0 and a high quality annotation is publicly available at https://valleyoak.ucla.edu/genomic-resources/  

In sum, I think the paper needs only minor revisions.  I encourage the authors to consider some synthetic statements at the end of each section and especially in the last section. Nonetheless, it is valuable that they have summarized so thoroughly the oak literature that many researchers and students will appreciate.

Author Response

Reviewer #2

Comments and Suggestions for Authors

This manuscript provides an excellent and thorough summary of the literature on stress response in oaks.  Researchers interested in oaks will find it very useful. The main sections of the paper correspond to organizational levels and each one is very thorough.  Typically, reviews will try to provide sub-conclusions within each section, and this paper does not, which is a minor limitation of the review.  

Firstly, we really appreciate all the helpful comments and suggestions of the three reviewers. In addition, we apologize to the reviewers because we think that they have not reviewed the English edited version. All the answers have been included in bold.

The authors use a few major terms without defining them.  The terms are:

1) Central Dogma of Molecular Biology--I did not know there was a central dogma.  Please define the central dogma and cite a reference.  Without the definition, it is not clear how use of the term adds to the organization of the paper.

Answer: We have modified the paragraph according to the comments of the reviewer, included the definition and some cites (line 110 to 112). Plants genetic variability can be analyzed following the biological information according to the Central Dogma of Molecular Biology and the omic cascade in eukaryotes, as can be observed in Q. ilex (Figure 1). Central Dogma was enunciated by Crick (Crick, 1970) and defined the relation between DNA, RNA and proteins, but nowadays different disciplines are interpreted their own “Central Dogma” including the relation with the omics cascade (Di leo et al., 2007; Patti et al., 2012; Fu et al., 2014; Srivastava, 2019) to the final output, the phenotype.

Crick, F. Central dogma of molecular biology. Nature 227, 561–563 (1970)

Di Leo A, Claudino W, Colangiuli D, Bessi S, Pestrin M, Biganzoli L. New strategies to identify molecular markers predicting chemotherapy activity and toxicity in breast cancer. Ann Oncol. Suppl 12:xii8-14 (2007)

Patti G, Yanes O & Siuzdak G. Metabolomics: the apogee of the omics trilogy. Nat Rev Mol Cell Biol 13, 263–269 (2012)

Fu Y, Dominissini D, Rechavi G. et al. Gene expression regulation mediated through reversible m6A RNA methylation. Nat Rev Genet 15, 293–306 (2014)

Srivastava S. Emerging Insights into the Metabolic Alterations in Aging Using Metabolomics. Metabolites, 9(12): 301 (2019)

2) Epigenetic priming--another term that the authors don't define. 

Answer: Epigenetic priming has been clarified in the main text. “Recently, the existence of epigenetic priming, or individual stress memory triggering natural plant defenses [Espinas et al., 2016; Balao et al., 2018], has been described in forest trees (lines 353-354)”

3). Systems Biology.  How do the authors define this term?  Cite a reference.  Be more specific on what this approach will reveal about stress response.  

Answer: We have modified the paragraph according to the comments of the reviewer and included the definition (line 644-645). “System biology is the study of biological systems whose behavior cannot be reduced to the linear sum of the functions of their parts (Palsson, 2015). For this purpose, computational modelling of molecular systems and an integrative interpretation of the different omics layers (such as transcriptomics, proteomics or metabolomics) are used (Breitling, 2010). This approach can potentially provide a more complete picture of the stress response in plants by exposing the interconnection between the different molecular levels and providing a global vision of the stress.”

Palsson, B. Systems biology. Constraint-based reconstruction and analysis. Cambridge University Press (2015)

Breitling R. What is systems biology? Frontiers in physiology, 1, 9. (2010)

4) I find the word "omics" to be jargon.  The title does not need it. In fact the title could be simplified to "Molecular and genomic research on stress response in the tree genus Quercus". 

Answer: We would say that the word “-omics” is widely accepted in the scientific community, as it is used in prestigious journals such as Nature (e.g. Palsson, 2002 or Zhu et al., 2020), and medical dictionaries (e.g. The American Heritage® Medical Dictionary, Segen's Medical Dictionary, or McGraw-Hill Concise Dictionary of Modern Medicine).

In this new version, we merged the Central Dogma (Genome->Transcriptome->Proteome) with the omics cascade which represents the relationships and interrelationships of this 3 major groups of biomolecules of the Dogma with the new layer metabolome (genomics, transcriptomics, proteomics and metabolomics). For this reason, we think that is necessary to keep the title as its current form.

Palsson, BO. In silico biology through ‘omics’. Nature Biotechnoly 20, 649–650 (2002)

Zhu C, Preissl S & Ren B. Single-cell multimodal omics: the power of many. Nature Methods 17, 11–14 (2020)

The American Heritage® Medical Dictionary Copyright © 2007, 2004 by Houghton Mifflin Company. Published by Houghton Mifflin Company.

Segen's Medical Dictionary. © 2012 Farlex, Inc.

McGraw-Hill Concise Dictionary of Modern Medicine. © 2002 by The McGraw-Hill Companies, Inc.

The section on epigenetics might benefit by some definitions on epigenetics from the broader literature. The authors might consider using background for 1 or 2 of these papers (but not all):

Bewick, A. J., Ji, L., Niederhuth, C. E., Willing, E.-M., Hofmeister, B. T., Shi, X., . . . Schmitz, R. J. (2016). On the origin and evolutionary consequences of gene body DNA methylation. Proceedings of the National Academy of Sciences, 113(32), 9111-9116. doi:10.1073/pnas.1604666113

Law, J. A., & Jacobsen, S. E. (2009). Dynamic DNA Methylation. Science, 323(5921), 1568-1569.

Bewick, A. J., & Schmitz, R. J. (2017). Gene body DNA methylation in plants. Current Opinion in Plant Biology, 36, 103-110. doi:10.1016/j.pbi.2016.12.007

Johannes, F., & Schmitz, R. J. (2019). Spontaneous epimutations in plants. New Phytologist, 221(3), 1253-1259. doi:10.1111/nph.15434

Niederhuth, C. E., Bewick, A. J., Ji, L., Alabady, M. S., Do Kim, K., Li, Q., . . . Udall, J. A. (2016). Genome Biology, 17(1), 194.

Niederhuth, C. E., & Schmitz, R. J. (2014). Molecular Plant, 7(3), 472-480. doi:DOI 10.1093/mp/sst165

Answer: We appreciate this comment and the list of references recommended by R2. We have included helpful definitions in the epigenetic section that will help readers to follow the review more easily. In addition, we have included a recent review (Amaral et al., 2020) where epigenetic processes in trees are detailed.

Conclusions and Future Perspectives: 

I find this section to be the weakest part of the paper.  Because the paper is a summary of various kinds of research rather than an exploration of specific questions or hypotheses, it is difficult for the authors to draw conclusions.  This section could be shortened to one paragraph with a few conclusions, which the authors would need to develop, and a second paragraph on future perspectives.  The authors are vague on this as well so I encourage them to think about what the questions are and what the approach should be.  I do not find their current ideas compelling.  

Answer: The final conclusion was rewritten following the reviewers' recommendations.

Minor comments:

The paper has scattered typos throughout the manuscript and inconsistent use of italics for species names in the text and especially literature cited.  I do not list fully, but below are a few I caught as examples. They typically would escape spell check functions.

We apologize for this. We suspect that reviewers have reviewed the version before MDPI's English editing service.

Line 199: change "this" to "these"

Answer: Done.

Line 278:  change "on" to "one"

Answer: Done.

One more paper that the authors might include in the epigenetic section is: 

Browne, L., Mead, A., Horn, C., Chang, K., A. Celikkol, Z., L. Henriquez, C., . . . Sork, V. L. (2020). Experimental DNA demethylation associates with changes in growth and gene expression of oak tree seedlings. G3: Genes|Genomes|Genetics, 10(3), 1019-1028. doi:10.1534/g3.119.400770

Answer: This paper has been included in the epigenetic section (lines 341).

In Table 1, the authors cite two papers on Q. lobata for climate adaptation.  They may also want to include: 

Browne, L., Wright, J. W., Fitz-Gibbon, S., Gugger, P. F., & Sork, V. L. (2019). Adaptational lag to temperature in valley oak (Quercus lobata) can be mitigated by genome-informed assisted gene flow. Proceedings of the National Academy of Sciences, 116(50), 25179-25185. doi:10.1073/pnas.1908771116

Answer: We have included the paper in DNA based markers and in Table 1.

In Table 1, the authors list valley oak genome V. 1 (Sork et al. 2016). The valley oak genome 3.0 and a high-quality annotation is publicly available at https://valleyoak.ucla.edu/genomic-resources/

Answer: Thank you so much for this comment. However, the valley oak genome 3.0 was not included in the review because we did not find the paper to be cited. Following this comment, we have included the link where is available as “Information on the genome of Q. lobata was published at later [40], being currently the version 3.0 of valley oak genome available at https://valleyoak.ucla.edu/genomic-resources/“.

In sum, I think the paper needs only minor revisions.  I encourage the authors to consider some synthetic statements at the end of each section and especially in the last section. Nonetheless, it is valuable that they have summarized so thoroughly the oak literature that many researchers and students will appreciate.

Answer: Synthetic statements have been included in each section of the review as well as the section “conclusions and future perspective” have been completely rewritten.

Reviewer 3 Report

The manuscript returns a current view (last 5 years) on the state of knowledge of molecular stress responses in the genus Quercus based on analyses of classical biochemistry, DNA based markers, genomics, epigenomics, transcriptomics, proteomics, and metabolomics.
The information given is detailed and precise. Generally, the text is readable, some chapters have language issues that merit some attention as to improve the understanding and clearness of the underlying messages for the reader. This specifically concerns the chapters Transciptomics, Proteomics, and
Metabolomics.

The major point to address, from my point of view, and which has led me to conclude that the manuscript needs major revision :
Whereas the amount of information is ample, detailed and well described, what seems necessary to me here is a stronger condensation of the knowledge as to learn from clearer and more valuable conclusions and perspectives on the genus and its relation to stress.

Some examples, that are relevant to the stress topic addressed in this
manuscript:
* Biotic and abiotic stressors can considerably vary in impact as in relation
to the age/ontogeny of the plant. Seedlings often tending to be more vulnerable than mature individuals. Further, controlled studies as in greenhouses and chambers usually work with young and potted plants, whereas in natura studies often focus on more mature individuals. Moreover stress in controlled conditions underlie less fluctuation as compared to field studies. This is bound to be reflected in transcriptomics, proteomics, and metabolomics etc. of the individual in relation to stress. It would be helpful to know where the research on Quercus currently is in these terms. This would allow to better position the state of the art in Quercus as compared to the other species and genus.

* The plant organs analysed by omics (e.g. leaf, root, fruit, or more being
general : above-/belowground) are of interest too, as omics will be different.
This has partly been focused on in this manuscript, but its hard to get a
general view from what is stated. An insight on the comparison of deciduous vs evergreen oak species would also be of interest in this context.

* The geographical occurance/origin of the species of Quercus spreads over
many latitudes and elevations, thus biomes/ climate zones, with some being
more prone/exposed to abiotic and biotic stress than others. Giving a view on
where the current research has been focusing on and where this is less the
case, would help to get a better global idea, in where we are in understanding
this genus.

* Concerning stresses, in future scenarios these usually occur in combination
(e.g. T rise & drought in mid latitudes, or increase in humidity and plant
pathogens in northern latitudes). It has been stated in this manuscript that
quite some studies exist on drought in Quercus, and also that combination of
stresses are more being studied (line 615). If the perspective of identifying
'elite' species or provenances is put forward in this manuscript, has
reviewing these studies helped to identify specific need(s) for knowledge?

In the chapter 10, addressing above points could help to structure the
messages to retain, and to incorporate motivating/promising information, on the progress that has been made. What I read between the lines is that science is in a frustrating position (lines 600-611), and that the proposed remedy would be to analyse many more genomes (line 622) and many individuals (line 630), however, contradicting statements are given on the potential cost-benefit relations of such an exercise (e.g. line 623 vs. 631ff). This chapter necessarily needs to emphasise on the genus Quercus, however, what could be learnt from other more strongly studied (woody) species or genus, as to better understand the genus Quercus.

Minor points

line 64 "it" --> "is"
line 55 "lost", what is mean there?
line 66 "afforded" --> "explained"
line 80ff. not only drought but also increased wetness/humidity in northern latitudes is predicted
line 125 --> change to : .... most of them BEING focused...
line 212 THESE have been ...
line 219 a set OF
line 220 In Q.aquifolioides patterns of 381 SNPs from 65 candidate genes WERE ANALYSED IN RELATION TO adaptation
line 224 ... have ALSO been reported ...
line 227f FURTHER, SNP markers HAVE been used...
line 252f ...which was selected FOR its small genome size, NO reference genome
sequence WAS AVAILABLE in forest species. THEREAFTER, the list ...
line 256 skip "already"
line 260 Rephrase positively : e.g. Information on the genome of Q.lobata was
published at later [40]
line 262 Q.suber WAS DETERMINED TO BE 1.45.
line 263ff : Rephrase e.g Progress on the Q.ilex genome is ongoing, datasets
have been ...
line 298 skip "a" ... been studied in few Quercus
line 319ff skip "We have ...".
line 385ff Unclear, rephrase sentence "This may be ... drought."
line 392 Unclear "possible drought-associated species-species adaptation
genes"
line 394 leaf --> LEAVES
line 397 Unclear "...was key under...", maybe "... was of key importance
under..."?
line 398 plural : "...there ARE species-specific RESPONSES..."
line 399 "... were THE ROS scavenging MACHINERY, ..."
line 399 replace "and" by comma ","
line 405 unclear : "about" --> "of" ?
line 407 rephrase "... considered as CANDIDATE housekeeping GENES, could
improve normalization of RT-qPCR over classical housekeeping GENES." (also
see line 410).
line 413 correct to "300 ppb"
line 444ff unclear and rephrase "Most proteomic studies ON the responses to
stresses have been performed in Q. ilex, most BEING limited to the use of the
gel-based" (??what??) "coupled to MALDI-TOF MS, and to a lesser 446 extent,
shotgun (LC-MSMS) platforms."
line 449 SYNDROMES
line 450 redundance "were focused ON the variation"
line 448 skip "already"
line 448 rephrase "... many of them being focused on the response ..."
line 469 REDUCED
line 482 "and for the genus Quercus in particular, () which..."
line 291 "... layer THE CLOSEST to THE phenotype, ..."
line 494 unclear "tolerant biomarkers" biomarkers for tolerance, or tolerant
to what?
line 495 unclear "traceability of acorns", rather "identification"?
line 505 underlying different stress conditions. (skip have been carried
..genus)
line 511-514 unclear, contradictory : widespread vs. unique?
line 534 unclear : being --> remaining ?
line 535 "its" refers to what?
line 535 In this line --> Similarly/ In analogy/ In this context
line 547-549 rephrase
line 549 coupleD
line 550 is most COMMON, and PERMITS the analysis
line 554 little size --> small
line 558f unclear, what is the message?
line 612-627
line 620 --> It is a long journey AZ TO ACHIEVE these goals. AS previously ...
line 631-633 message unclear, rephrase

The use of the word "also" is excessive, try to avoid this where possible.

Author Response

Reviewer #3

Comments and Suggestions for Authors

The manuscript returns a current view (last 5 years) on the state of knowledge of molecular stress responses in the genus Quercus based on analyses of classical biochemistry, DNA based markers, genomics, epigenomics, transcriptomics, proteomics, and metabolomics. The information given is detailed and precise. Generally, the text is readable, some chapters have language issues that merit some attention as to improve the understanding and clearness of the underlying messages for the reader. This specifically concerns the chapters Transcriptomics, Proteomics, and Metabolomics.

Firstly, we really appreciate all the helpful comments and suggestions of the three reviewers. In addition, we apologize to the reviewers because we think that they have not reviewed the English edited version. All the answers have been included in bold.

The major point to address, from my point of view, and which has led me to conclude that the manuscript needs major revision:

Whereas the amount of information is ample, detailed and well described, what seems necessary to me here is a stronger condensation of the knowledge as to learn from clearer and more valuable conclusions and perspectives on the genus and its relation to stress.

We understand the reviewer's point of view, where the search for conclusions at the biological level is the ultimate goal of a study. We have rewrite the manuscript with this in mint. In the genus Quercus, the molecular analysis of the stress response has been initiated with a partial and segmented view of the different processes. At the molecular level, we are at the beginning of this objective, that is to say, "we are beginning to have frames of the film", which will help us in future years to complete the film. For example, in our experimental system Q. ilex, one of the most advanced species at the molecular level within the genus and in which we have enough experience, is beginning to respond to developmental processes in acorns, or at the level of response to stress with genes or gene products related to resilience and tolerance to the decline syndrome (mainly drought and/or Phytophthora cinnamomi). This initial state is common in forest species, where more progress has been made in the face of abiotic stress with recent reviews on the molecular basis (Estravis-Barcala et al., 2020; Harfouche et al., 2014). At level of species, the model in trees, Populus, has been the most advanced with numerous reviews at the molecular level: some examples (Wullschleger et al., 2009; Li & He, 2019; Melnikova, et al., 2017; Chen & Polle, 2010; Wullschleger et al., 2009; Polle & Chen, 2015). By contrast, in genus Quercus this is first review at all molecular level.

Harfouche A, Meilan R, Altman A. Molecular and physiological responses to abiotic stress in forest trees and their relevance to tree improvement. Tree Physiology. 34(11): 1181-98 (2014)

Estravis-Barcala M, Mattera MG, Soliani C, Bellora N, Opgenoorth L, Heer, K, & Arana MV. Molecular bases of responses to abiotic stress in trees. Journal of experimental botany, 71(13), 3765–3779 (2020)

Li, A., & He, W. Molecular Aspects of an Emerging Poplar Canker Caused by Lonsdalea populi. Frontiers in microbiology, 10, 2496 (2019)

Melnikova NV, Borkhert EV, Snezhkina, AV, Kudryavtseva AV, & Dmitriev AA. Sex-Specific Response to Stress in Populus. Frontiers in plant science, 8, 1827 (2017)

Chen S, & Polle A. Salinity tolerance of Populus. Plant biology (Stuttgart, Germany), 12(2), 317–333 (2010)

Wullschleger ST, Weston DJ & Davis JM. Populus Responses to Edaphic and Climatic Cues: Emerging Evidence from Systems Biology Research, Critical Reviews in Plant Sciences, 28:5, 368-374 (2009)

Polle A & Chen S. On the salty side of life: molecular, physiological and anatomical adaptation and acclimation of trees to extreme habitats. Plant, cell & environment, 38(9), 1794–1816 (2015)

Some examples, that are relevant to the stress topic addressed in this manuscript:

* Biotic and abiotic stressors can considerably vary in impact as in relation
to the age/ontogeny of the plant. Seedlings often tend to be more vulnerable than mature individuals. Further, controlled studies as in greenhouses and chambers usually work with young and potted plants, whereas in natura studies often focus on more mature individuals. Moreover, stress in controlled conditions underlies less fluctuation as compared to field studies. This is bound to be reflected in transcriptomics, proteomics, and metabolomics etc. of the individual in relation to stress. It would be helpful to know where the research on Quercus currently is in these terms. This would allow to better position the state of the art in Quercus as compared to the other species and genus.

Answer: After reviewing, in details, all the studies included in each section, we have added specific information regarding to the developmental stage of these species as well as the conditions where the experiments were carried out.

* The plant organs analysed by omics (e.g. leaf, root, fruit, or more being general: above-/belowground) are of interest too, as omics will be different. This has partly been focused on in this manuscript, but it’s hard to get a general view from what is stated. An insight on the comparison of deciduous vs evergreen oak species would also be of interest in this context.

Answer: The plant organs analysed by omics have been included in all studies mentioned in the review. We have also specified if the Quercus spp. is deciduous or evergreen showing that the study in deciduous species is more extended in the different omics than evergreen species, except for the species Q. ilex and Q. suber, which are very advanced in their molecular study. For this, we have included a new column in Table 1.

* The geographical occurance/origin of the species of Quercus spreads over many latitudes and elevations, thus biomes/ climate zones, with some being more prone/exposed to abiotic and biotic stress than others. Giving a view on where the current research has been focusing on and where this is less the case, would help to get a better global idea, in where we are in understanding this genus.

Answer: In Quercus the knowledge is very fragmented without a clear establishment of provenances/origin or clear phenotypes more adapted to biotic or abiotic stress. Some studies based on provenances more adapted to one type of stress have been done on the genus, but in general is not very advanced. Only transcriptomic studies in Q. lobata provenances (Gugger etal., 2017; Mead et al., 2019) or proteomics ones in Q. ilex (Jorge et al., 2006; Valero-Galvan et al., 2013; Shaier-Hammami et al., 2013) were addressed on those geographical regions affected by specific stress. Interesting results were obtained in these studies shedding light on the molecular mechanisms activated which varies among different populations studied. We think it is necessary to build consensus reference populations for each species in all countries, as well as establishing defined phenotypes for comparison and integration. This idea has been included in the last rewritten section.

* Concerning stresses, in future scenarios these usually occur in combination (e.g. T rise & drought in mid latitudes, or increase in humidity and plant pathogens in northern latitudes). It has been stated in this manuscript that quite some studies exist on drought in Quercus, and also that combination of stresses are more being studied (line 615). If the perspective of identifying 'elite' species or provenances is put forward in this manuscript, has reviewing these studies helped to identify specific need(s) for knowledge?

Answer: We hope to understand and answer correctly the comment made by the referee. We think that the knowledge of combined stresses will help to understand properly the mechanisms of response to biotic and abiotic stresses. Initially, the tendency was to determine the effect of a single stress to conclude biological changes in the individual. However, now, the tendency is to simulate the natural conditions with the application of combined stresses, being common to determine the effect of two stresses due to the complexity of the experimental design. These studies in combination with studies under controlled conditions will constitute a powerful approach to determine the response to multiple stresses. In the manuscript, we have added more information related to response to biotic and abiotic stresses in those studies where several stresses were combined.

The use all the molecular information collected in this review, although still scarce, will undoubtedly help on the identification of those elite genotypes that, after clonal propagation can be used in reforestation programs under a climate change scenario. Even so, as far as Q. ilex research is concerned, the real advance will come with genome sequencing, which will allow the validation of the data generated so far using omics tools.

In the chapter 10, addressing above points could help to structure the messages to retain, and to incorporate motivating/promising information, on the progress that has been made. What I read between the lines is that science is in a frustrating position (lines 600-611), and that the proposed remedy would be to analyse many more genomes (line 622) and many individuals (line 630), however, contradicting statements are given on the potential cost-benefit relations of such an exercise (e.g. line 623 vs. 631ff). This chapter necessarily needs to emphasise on the genus Quercus, however, what could be learnt from other more strongly studied (woody) species or genus, as to better understand the genus Quercus.

Answer: We really appreciate these comments and the “conclusion and future perspective” has been completely modified.

Minor points

line 64 "it" --> "is"

Answer: Done

line 55 "lost", what is mean there?
Answer: We meant loss rather than lost.

line 66 "afforded" --> "explained"
Answer: Done

line 80ff. not only drought but also increased wetness/humidity in northern latitudes is predicted
Answer: Modified to “Simulation models predict an increase in both temperature and the frequency of severe drought episodes, or in northern latitudes the increased humidity, which will determine forest vulnerability.

line 125 --> change to: .... most of them BEING focused...
Answer: Done

line 212 THESE have been ...
Answer: Done

line 219 a set OF
Answer: Done

line 220 In Q. aquifolioides patterns of 381 SNPs from 65 candidate genes WERE ANALYSED IN RELATION TO adaptation

Answer: It has been changed in the main text.

line 224 ... have ALSO been reported ...
Answer: Done

line 227f FURTHER, SNP markers HAVE been used...
Answer: Modified

line 252f ...which was selected FOR its small genome size, NO reference genome sequence WAS AVAILABLE in forest species. THEREAFTER, the list ...
Answer: Thank you a lot. It has been modified following your suggestion.

line 256 skip "already"
Answer: Removed

line 260 Rephrase positively: e.g. Information on the genome of Q. lobata was published at later [40]

Answer: Modified

line 262 Q. suber WAS DETERMINED TO BE 1.45.
Answer: Changed

line 263ff : Rephrase e.g Progress on the Q.ilex genome is ongoing, datasets have been ... Answer: This has been modified following your suggestion.

line 298 skip "a" ... been studied in few Quercus
Answer: Done

line 319ff skip "We have ...".
Answer: Modified

line 385ff Unclear, rephrase sentence "This may be ... drought."

Answer: Modified to “This may be due to the fact that the response pathways are different among populations adapted to combined stress than those that are usually only under one stress (drought)”.

line 392 Unclear "possible drought-associated species-species adaptation genes"

Answer: Modified in the English version to “possible drought-associated species-specific adaptation genes”. The sentence is in green in the track manuscript as it has already been modified in the revised English version.

line 394 leaf --> LEAVES

Answer: Done

line 397 Unclear "...was key under...", maybe "... was of key importance under..."?

Answer: Modified

line 398 plural: "...there ARE species-specific RESPONSES..."
Answer: Done

line 399 "... were THE ROS scavenging MACHINERY, ..."
Answer: Done

line 399 replace "and" by comma ","
Answer: Modified

line 405 unclear: "about" --> "of" ?
Answer: Done

line 407 rephrase "... considered as CANDIDATE housekeeping GENES, could improve normalization of RT-qPCR over classical housekeeping GENES." (also see line 410).

Answer: It has been modified as reviewer proposed.

line 413 correct to "300 ppb"
Answer: Changed.

line 444ff unclear and rephrase "Most proteomic studies ON the responses to stresses have been performed in Q. ilex, most BEING limited to the use of the gel-based" (??what??) "coupled to MALDI-TOF MS, and to a lesser 446 extent, shotgun (LC-MSMS) platforms."

Answer: We have clarified that sentence to “Most proteomic studies on the responses to stresses have been performed in Q. ilex, most being limited to the use of the gel-based (one or two-dimensional gel electrophoresis) coupled to MALDI-TOF MS (Matrix-Assisted Laser Desorption/Ionization time-of-flight mass spectrometry), and to a lesser extent, shotgun gel-free (LC-MSMS) platforms”.

line 449 SYNDROMES
Answer: Done

line 450 redundance "were focused ON the variation"
Answer: Done

line 448 skip "already"
Answer: Done

line 448 rephrase "... many of them being focused on the response ..."
Answer: Modified to “with many research in abiotic and biotic stresses related to decline syndromes (drought and P. cinnamomi infestation).

line 469 REDUCED
Answer: Done

line 482 "and for the genus Quercus in particular, () which..."
Answer: Done

line 291 "... layer THE CLOSEST to THE phenotype, ..."
Answer: Done

line 494 unclear "tolerant biomarkers" biomarkers for tolerance, or tolerant to what?

Answer: Modified to “biomarkers for stress tolerance”.

line 495 unclear "traceability of acorns", rather "identification"?

Answer: We have clarified it. We meant that nutraceutical and bioactive acorn compounds can be identified by metabolomics. These compounds may confer an important economic value upon holm oak and may signify a change in perception of the plant from an environmental to a productivity perspective.

line 505 underlying different stress conditions. (skip have been carried...genus)
Answer: Removed.

line 511-514 unclear, contradictory: widespread vs. unique?

Answer: Modified to “In trees, the emission of isoprenoids is widespread, with the genus Quercus being unique in evolved species emitting isoprene, monoterpene, or non-isoprenoids.”

line 534 unclear: being --> remaining ?

Answer: Done

line 535 "its" refers to what?
Answer: It has been modified.

line 535 In this line --> Similarly/ In analogy/ In this context
Answer: Modified by “In this context”.

line 547-549 rephrase.

Answer: Modified by “In recent years, the advent of massive data analysis has allowed metabolomic analysis from a global perspective, which has permitted a more in-depth understanding of the metabolome in the presence of different stresses”. The sentence is in green in the track manuscript as it has already been modified in the revised English version.

line 549 coupleD
Answer: Done

line 550 is most COMMON, and PERMITS the analysis
Answer: Modified

line 554 little size --> small
Answer: Done

line 558f unclear, what is the message?

Answer: Modified to “The root metabolism under drought was analyzed in Q. rubra and Q. alba and it was re-ported that the amount of lignins was not modified in the fine roots, although it did increase in others part of the plants as in leaves”.

line 620 --> It is a long journey AZ TO ACHIEVE these goals. AS previously ...
Answer: Modified.

line 631-633 message unclear, rephrase

Answer: This sentence has been clarified.

The use of the word "also" is excessive, try to avoid this where possible.

Answer: We have reviewed the manuscript to remove “also” in those sentences where it is possible.

Round 2

Reviewer 1 Report

The authors made the reformulations proposed.

There are only a small thing that I would appreciate if the authors took in consideration:

The reference Rey et al. 2020  is not appropriate and I suspect that the authors wanted to refer the other work of the team:

Rey MD, Castillejo MA, Sanchez-Lucas R, Guerrero-Sanchez VM, Lopez-Hidalgo C, Romero- Rodriguez C et al (2019) Proteomics, Holm oak (Quercus ilex L.) and other recalcitrant and orphan forest tree species: how do they see each other? Int J Mol Sci 20:692. https://doi.org/10. 3390/ijms20030692

 Please alter the reference accordingly

Also in line 301 there is a mistake: The genome size of Q. ilex, calculated by flow cytometry, was shown to be 1.87 Gb/12C  Please include calculated and alter 12C by 2C

Author Response

Answer: We are really grateful for your comment. We agree by R1, there is a mistake in Rey et al., 2019 (citation number 123). We have modified it to “Rey, M. D., Castillejo, M. Á., Sánchez-Lucas, R., Guerrero-Sanchez, V. M., López-Hidalgo, C., Romero-Rodríguez, C., ... & Jorrín-Novo, J. V. (2019). Proteomics, Holm Oak (Quercus ilex L.) and Other Recalcitrant and Orphan Forest Tree Species: How do They See Each Other?. International journal of molecular sciences20(3), 692” rather than “Rey, M.D.; Valledor, L.; Castillejo, M.A.; Sánchez-Lucas, R.; López-Hidalgo, C.; Guerrero-Sánchez, V.M.; Colina, F.J.; 1077 Escandón, M.; Maldonado-Alconada, A.M.; Jorrín-Novo, J. V Recent advances in MS-based plant proteomics: proteomics 1078 data validation through integration with other classic and -omics approaches. In Progress in Botany, vol 81; Cánovas, F., Lüttge, 1079 U., Leuschner, C., Risueño, M., Eds.; Springer, Cham., 2019 ISBN 978-3-030-36327-7.”

Also in line 301 there is a mistake: The genome size of Q. ilex, calculated by flow cytometry, was shown to be 1.87 Gb/12C. Please include calculated and alter 12C by 2C

Answer: We have already included it and 12C has been modified by 2C.

“The estimated genome size of Q. ilex, by flow cytometry, was approximately 1860 Mb/2C with a total length of 1.87 Gb/2C and, as expected, it is formed by 12 pairs of chromosomes [120,123–125]).”

Reviewer 2 Report

The manuscript has been revised satisfactorily.  I am satisfied with their changes and thank them for producing this manuscript, which will be helpful to many of us in the oak research community and others who study trees.

Author Response

Answer: We sincerely appreciate all valuable comments and suggestions, which helped us to improve the quality of the review.